# Megalencephalic leukoencephalopathy with subcortical cysts is a developmental disorder of the gliovascular unit

Alice Gilbert[1,2], Xabier Elorza-Vidal[1], Armelle Rancillac[3], Audrey Chagnot[4], Mervé Yetim[4], Vincent Hingot[5], Thomas Deffieux[5], Anne-Cécile Boulay[1], Rodrigo Alvear-Perez[1], Salvatore Cisternino[6], Sabrina Martin[7], Sonia Taïb[7], Aontoinette Gelot[8], Virginie Mignon[9], Maryline Favier[10], Isabelle Brunet[7], Xavier Declèves[6,11], Mickael Tanter[5], Raul Estevez[12,13], Denis Vivien[4], Bruno Saubaméa[6,9], Martine Cohen-Salmon[1]*

[1]Physiology and Physiopathology of the Gliovascular Unit Research Group, Center for Interdisciplinary Research in Biology (CIRB), College de France, CNRS Research in Biology (CIRB), College de France, CNRS, Paris, France; [2]École doctorale Cerveau Cognition Comportement "ED3C" N°158, Pierre and Marie Curie University, Paris, France; [3]Neuroglial Interactions in Cerebral Physiopathology Research Group, Center for Interdisciplinary Research in Biology (CIRB), College de France, Labex Memolife, Université PSL, Paris, France; [4]Normandie University, UNICAEN, INSERM, GIP Cyceron, Institut Blood and Brain, Physiopathology and Imaging of Neurological Disorders, Caen, France; [5]Physics for Medicine Paris, ESPCI Paris, PSL University, Paris, France; [6]Université de Paris, Faculté de Santé, Paris, France; [7]Molecular Control of the Neurovascular Development Research Group, Center for Interdisciplinary Research in Biology (CIRB), College de France, Labex Memolife, Université PSL, Paris, France; [8]Service d'anatomie et cytologie pathologie de l'hôpital Armand Trousseau, Paris, France; [9]Cellular and Molecular Imaging Facility, US25 INSERM, UMS3612 CNRS, Faculty of Pharmacy, University of Paris, Paris, France; [10]Plateforme HistIM Institut Cochin, Paris, France; [11]Biologie du médicament et toxicologie, Assistance Publique – hôpitaux de Paris, APHP, Hôpital Cochin, Paris, France; [12]Unitat de Fisiología, Departament de Ciències Fisiològiques, IDIBELL-Institute of Neurosciences, Universitat de Barcelona, L'Hospitalet de Llobregat, Barcelona, Spain; [13]Centro de Investigación en Red de Enfermedades Raras (CIBERER), Barcelona, Spain

*For correspondence:
martine.cohen-salmon@college-de-france.fr

Competing interest: The authors declare that no competing interests exist.

**Abstract** Absence of the astrocyte-specific membrane protein MLC1 is responsible for megalencephalic leukoencephalopathy with subcortical cysts (MLC), a rare type of leukodystrophy characterized by early-onset macrocephaly and progressive white matter vacuolation that lead to ataxia, spasticity, and cognitive decline. During postnatal development (from P5 to P15 in the mouse), MLC1 forms a membrane complex with GlialCAM (another astrocytic transmembrane protein) at the junctions between perivascular astrocytic processes. Perivascular astrocytic processes along with blood vessels form the gliovascular unit. It was not previously known how MLC1 influences the physiology of the gliovascular unit. Here, using the *Mlc1* knock-out mouse model of MLC, we demonstrated that MLC1 controls the postnatal development and organization of perivascular astrocytic processes, vascular smooth muscle cell contractility, neurovascular coupling, and intraparenchymal interstitial fluid clearance. Our data suggest that MLC is a developmental disorder of the

gliovascular unit, and perivascular astrocytic processes and vascular smooth muscle cell maturation defects are primary events in the pathogenesis of MLC and therapeutic targets for this disease.

## Editor's evaluation

In this manuscript by Gilbert et al., the authors investigate how deletion of astrocytic membrane protein, MLC1, causes a rare disease called megalencephalic leukoencephalopathy with subcortical cysts (MLC). Through multiple experimental approaches, the authors show that Mlc1 knock-out mice exhibit defects in postnatal maturation of perivascular astrocyte coverage as well as vascular smooth muscle cell contractility, neurovascular coupling, and parenchymal CSF flow. Together, this work provides important frame works that MLC is caused by defects in the development of gliovascular unit.

## Introduction

Megalencephalic leukoencephalopathy with subcortical cysts (MLC) is a rare type of leukodystrophy (OMIM 604004) mainly caused by mutations in the *MLC1* gene (MIM #605908) (*Leegwater et al., 2001*; *Topçu et al., 2000*). Patients with MLC display early-onset macrocephaly and progressive white matter vacuolation, leading to slowly progressive ataxia, spasticity, and cognitive decline. Most mutations in *MLC1* result in the degradation of the encoded protein MLC1 (*Duarri et al., 2008*; *Lanciotti et al., 2016*; *Leegwater et al., 2001*), a membrane protein that is specifically expressed by the astrocytic lineage in the brain and present at high levels at the junctions between perivascular astrocytic processes (*Hoegg-Beiler et al., 2014*; *Wang et al., 2021*). At present, there is no cure for MLC, only symptomatic treatments and supportive care are available. Although the physiopathological mechanisms leading to MLC have not been characterized, the strong expression of MLC1 in perivascular astrocytic processes and other recent observations suggest that the protein has a role in gliovascular functions, particularly the regulation of ion/water homeostasis (*Capdevila-Nortes et al., 2013*; *Hoegg-Beiler et al., 2014*; *Ridder et al., 2011*). Indeed, MLC patients present widespread brain edema and swollen perivascular astrocytic processes (*Bugiani et al., 2017*; *Dubey et al., 2015*). GlialCAM, another transmembrane protein forming a complex with MLC1 and responsible for its endoplasmic reticulum exit, is an auxiliary subunit of ClC-2, an inward rectifier chloride channel expressed in a subtype of astrocytes (*Benesova et al., 2012*; *Bugiani et al., 2017*; *Estévez et al., 2018*; *Hoegg-Beiler et al., 2014*; *Jeworutzki et al., 2012*). Recessive and dominant mutations in GlialCAM cause MLC subtypes MLC2A and MLC2B, respectively (*van der Knaap et al., 1993*). In vitro, the MLC1/GlialCAM complex indirectly regulates other ion channels, such as TRPV4 and LRRC8 (*Elorza-Vidal et al., 2018*).

Despite the above observations, it is still not known whether and how MLC1 influences the physiology of the gliovascular unit, the functional interface comprising perivascular astrocytic processes and the brain vessels. We recently reported that MLC1 expression in mouse perivascular astrocytic processes starts around postnatal day (P) 5 and that the MLC1/GlialCAM complex forms progressively from P5 to P15, with the protein deposits creating a meshwork between the astrocytes' perivascular membranes (*Gilbert et al., 2019*). We also demonstrated that this postnatal period is a developmental window for the molecular maturation of brain endothelial cells, particularly with regard to their efflux properties and for the contractility of vascular smooth muscle cells (*Gilbert et al., 2019*; *Slaoui et al., 2021*). Given that astrocytes are key regulators of cerebrovascular development and function (e.g., blood–brain barrier integrity, immune quiescence and perivascular homeostasis, neurovascular coupling) (*Abbott et al., 2006*; *Alvarez et al., 2013*; *Castro Dias et al., 2019*; *Cohen-Salmon et al., 2021*), we hypothesized that the MLC1/GlialCAM complex might influence the postnatal differentiation of the vascular system.

Here, we characterized key aspects of the molecular and functional organization of the gliovascular unit in *Mlc1* knock-out (KO) mice, a preclinical model of MLC that recapitulates several important features of the disease and that can be used to examine the pathological cascade (*Hoegg-Beiler et al., 2014*). Our results revealed that MLC1 is a critical factor in the postnatal maturation and function of the gliovascular unit.

## Results

### The absence of MLC1 results in accumulation of fluid in the brain but does not alter blood–brain barrier integrity or the organization of the endothelial network

Astrocytes influence several properties of endothelial cells, such as blood–brain barrier integrity (*Abbott et al., 2006*; *Alvarez et al., 2013*; *Castro Dias et al., 2019*; *Cohen-Salmon et al., 2021*). We recently demonstrated that the postnatal maturation of the MLC1/GlialCAM complex in perivascular astrocytic processes from P5 to P15 coincides with the progressive increase in endothelium-specific proteins that contribute to blood–brain barrier integrity, such as the tight junction protein claudin5 and the endothelial luminal ATP-binding cassette (ABC) efflux transporter P-glycoprotein (P-gP), suggesting that perivascular astrocytic processes and the blood–brain barrier mature in parallel (*Gilbert et al., 2019*; *Slaoui et al., 2021*). We therefore investigated whether the absence of MLC1 in the *Mlc1* KO mouse influenced postnatal endothelial maturation. To that end, we used qPCRs to characterize the expression of *Abcb1* (encoding P-gP) and *Cldn5* (encoding claudin5) on P5, P15, and P60 in whole-brain microvessels isolated from WT and *Mlc1* KO animals (*Boulay et al., 2015*; *Figure 1A*, *Figure 1—source data 1*). In the WT, expression of both analyzed mRNAs on P5 and P15 confirmed the progressive postnatal molecular maturation of endothelial cells (*Figure 1A*). There were no significant differences between *Mlc1* KO mice and WT mice in this respect (*Figure 1A*). Consistently, the protein levels of P-gP and claudin5 in purified brain microvessels determined by western blots were also similar in *Mlc1* KO and WT mice at all stages (*Figure 1B*, *Figure 1—source data 1*). We next assessed the blood–brain barrier integrity in *Mlc1* KO and WT mice on P60. We first observed that the apparent diffusion coefficient (measured using magnetic resonance imaging) was higher in *Mlc1* KO mice (*Figure 1C*, *Figure 1—figure supplement 1*, *Figure 1—source data 1*, *Figure 1—figure supplement 1—source data 1*). Furthermore, the volume of the ventricles and the brain estimated from the anatomical T2-weighted magnetic resonance imaging acquisition was larger in *Mlc1* KO mice than in WT mice (*Figure 1—figure supplement 1*), although the relative ventricular volumes did not differ (*Figure 1—figure supplement 1*). These results reflected the previously described presence of fluid in the parenchyma in *Mlc1* KO mice (*Dubey et al., 2015*). However, this fluid accumulation was not related to leakage of the blood–brain barrier. Indeed, the vascular volume measured by in situ brain perfusion of [$^{14}$C] sucrose (a marker of vascular space and integrity; *Dagenais et al., 2000*) was the same in *Mlc1* KO and WT mice and so suggested that the blood–brain barrier was not leaky, even in the shear stress conditions (increased hydrostatic pressure: 180 mmHg) produced by the addition of human serum albumin to the perfusate (*Ezan et al., 2012*; *Figure 1D*, *Figure 1—source data 1*). Lastly, we assessed the endothelium architecture by analyzing vessel length, branching, and tortuosity in the whole cleared somatosensory cortex of WT and *Mlc1* KO on P60 immunolabeled for the endothelium-specific protein Pecam-1 (*Figure 1E and F*, *Figure 1—source data 1*). There were no differences between WT and *Mlc1* KO mice with regard to these architectural parameters in either the parenchymal or pial vasculature at the cortical surface (*Figure 1E and F*).

Hence, the absence of MLC1 leads to fluid accumulation in the brain but has no obvious impact on the postnatal molecular maturation of ECs, blood–brain barrier integrity, or endothelium architecture.

### MLC1 is crucial for contractile maturation of vascular smooth muscle cells, arterial perfusion, and neurovascular coupling

We recently showed that vascular smooth muscle cells mature postnatally in mice and humans, with the progressive acquisition of contractility (from P5 onward in mice and from birth in humans) and the extension of the vascular smooth muscle cell network (*Slaoui et al., 2021*). Here, we investigated the vascular smooth muscle cells' status in *Mlc1* KO mice during postnatal development. We first used qPCRs to compare the mRNA expression of *Acta2* (encoding smooth muscle actin [SMA]) in microvessels purified from WT and *Mlc1* KO whole brain on P5, P15, and P60 (*Boulay et al., 2015*; *Figure 2A*, *Figure 2—source data 1*). We also measured the mRNA expression of *Atp1b1* (a vascular smooth muscle cell-specific gene stably expressed during postnatal development) as a marker of vascular smooth muscle cell density in purified brain microvessels (*He et al., 2016*; *Vanlandewijck et al., 2018*). In WT mice, the level of *Acta2* mRNA rose progressively from P5 to P15 while the level of *Atp1b1* mRNA remained stable (*Figure 2A*). In *Mlc1* KO mice, however, *Acta2* was significantly

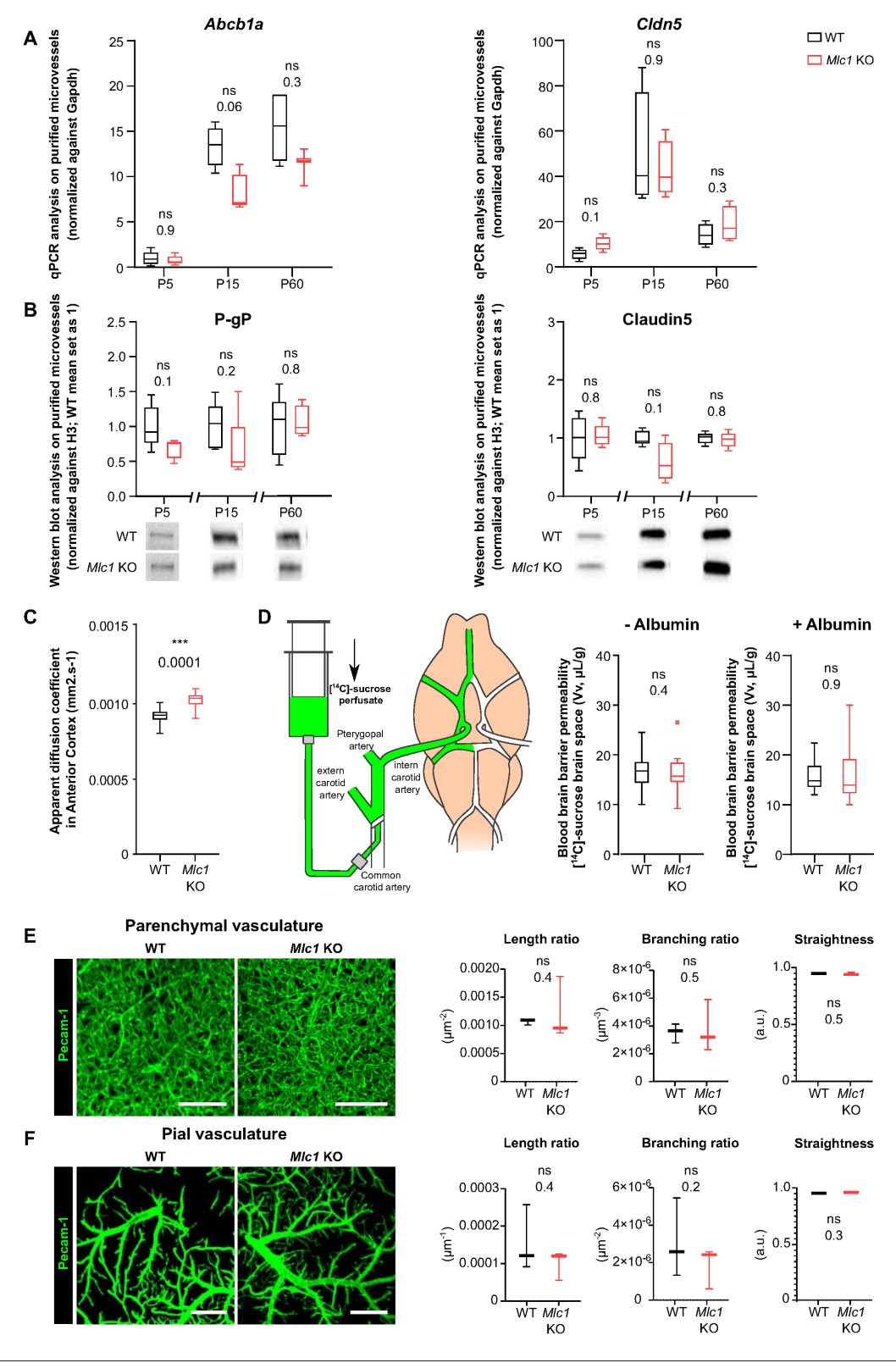

**Figure 1.** The absence of MLC1 has no effect on blood–brain barrier integrity or the organization of the endothelial network. (**A**) qPCR determination of mRNA expression of *Abcb1* (encoding P-gP) and *Cldn5* (encoding claudin5) in microvessels purified from WT and *Mlc1* KO whole brains on postnatal day (P)5, P15, and P60. Signals were normalized against *Gapdh*. Groups were compared using a two-tailed Mann–Whitney test. The data are

*Figure 1 continued on next page*

*Figure 1 continued*

presented as a Tukey box plot (n = 3 or 4 samples per genotype; number of brains pooled per sample: five for P5; three for P15; two for P60). (**B**) Western blot detection and analysis of P-gP and claudin5 in protein extracts from microvessels purified from WT and *Mlc1* KO whole brains on P5, P15, and P60. Signals were normalized against histone H3. Two-tailed Mann–Whitney test. The data are represented in a Tukey box plot (n = 4 or 5 samples per genotype; number of brains pooled per sample: five for P5; three for P15; two for P60). (**C**) Apparent diffusion coefficient values in the cortex of 2-month-old WT and *Mlc1* KO mice. Two-tailed Student's t-test. The data are represented in a Tukey box plot (n = 7 mice per genotype). (**D**) Blood–brain barrier integrity, assessed by measuring the brain vascular volume (Vv, in µL/g), after in situ brain perfusion with [$^{14}$C]-sucrose and a normal hydrostatic vascular pressure (without albumin [Albumin-]; 120 mmHg) or an elevated hydrostatic vascular pressure (with albumin [Albumin+]; 180 mmHg) in 2-month-old WT (in black) and *Mlc1* KO mice (in red). Two-tailed Mann–Whitney test. The data are represented in a Tukey box plot (n = 8 WT and 9 *Mlc1* KO mice for Albumin -; n = 11 WT and 12 *Mlc1* KO mice for Albumin+).(**E**) Representative 3D images of the endothelial architecture in cleared somatosensory cortex. Parenchymal (Z-stack 320 µm; scale bar: 100 µm) (top) and pial (bottom) vessels (Z-stack 50 µm; scale bar: 500 µm) samples from 2-month-old WT and *Mlc1* KO mice, after immunolabeling for Pecam1. (**F**) A comparative analysis of vessel length, branching, and tortuosity in WT mice (in black) and *Mlc1* KO mice (in red) in the parenchymal cortex (top), normalized on sample volume, and cortical surface (bottom), normalized on sample surface. One-tailed Mann–Whitney test. The data are represented in a Tukey box plot (n = 3 mice per genotype). The data are given in *Figure 1—source data 1*. *p≤0.05, **p≤0.01, ***p≤0.001, and ns: not significant.

The online version of this article includes the following figure supplement(s) for figure 1:

**Source data 1.** The absence of MLC1 has no effect on blood–brain barrier integrity or the organization of the endothelial network.

**Figure supplement 1.** The absence of MLC1 causes overall swelling of the brain.

**Figure supplement 1—source data 1.** The absence of MLC1 causes overall swelling of the brain.

---

downregulated on P5 and P15, while *Atp1b1* levels were unchanged (*Figure 2A*). We next analyzed the protein levels of SMA in microvessels purified from WT and *Mlc1* KO whole brain on P5, P15, and P60 by western blot (*Figure 2B*, *Figure 2—source data 1*). A small decrease in SMA expression was found in *Mlc1* KO microvessels from P15 onward, although it became significant only at P60. Moreover, additional bands of lower molecular weights resembling a degradation pattern were detected at this stage (*Figure 2B*). To determine whether the decrease in SMA expression and the putative increase in its degradation were related to vascular smooth muscle cell degeneration, we used an immunofluorescence assay to detect SMA on 2-month-old WT and *Mlc1* KO whole cleared somatosensory cortices (*Figure 2C and D*). No discontinuities in the labeling were detected in the parenchymal (*Figure 2C*) or pial vasculature (at the cortical surface) (*Figure 2D*). Moreover, the SMA-positive vessels' length, branching, tortuosity, and number of anastomoses (analyzed only in pial vessels) were the same in *Mlc1* KO and WT mice (*Figure 2C and D*). These findings indicate that the absence of MLC1 perturbs the developmental expression of SMA but does not affect the development of the vascular smooth muscle cell network.

To further assess the functional consequences of this molecular change, we compared the ex vivo contractility of vascular smooth muscle cells in brain slices obtained from *Mlc1* KO and WT mice on P5, P15, and P60 (*Figure 3A–C*, *Figure 3—source data 1*). We recorded the vasomotor changes in cortical arterioles upon exposure for 2 min to the thromboxane A$_2$ receptor agonist U46619 (9,11-dideoxy-11a,9a-epoxymethanoprostaglandin F2α, 5 nM), which acts directly on vascular smooth muscle cells to induce a reversible vasoconstriction. On P5, application of U46619 had a small effect on vessel diameter in both WT and *Mlc1* KO mice (*Figure 3B and C*). In contrast, a clear vasoconstriction was observed on P15 (*Figure 3B and C*). Strikingly, the amplitude and speed of vasoconstriction were significantly lower in *Mlc1* KO mice from P15 onward (*Figure 3B and C*, *Figure 3—source data 1*). These results indicate that the postnatal acquisition of contractility is impaired in the absence of MLC1.

Given this phenotype, we next hypothesized that arterial tonicity might be impaired in *Mlc1* KO mice. We addressed this question by performing ultrasound localization microscopy in vivo imaging to reveal the brain vasculature at a microscopic resolution after intravenous microbubble injection (*Figure 3D*). *Mlc1* KO mice displayed significantly lower blood perfusion, suggesting narrower penetrating arteries (*Figure 3D*, *Figure 3—source data 1*). Vasomotricity and cerebral blood flow are tightly coupled to neuronal energy demand, in a process referred to as neurovascular coupling or functional hyperemia (*Iadecola, 2017*). We then used functional ultrasound imaging to measure the

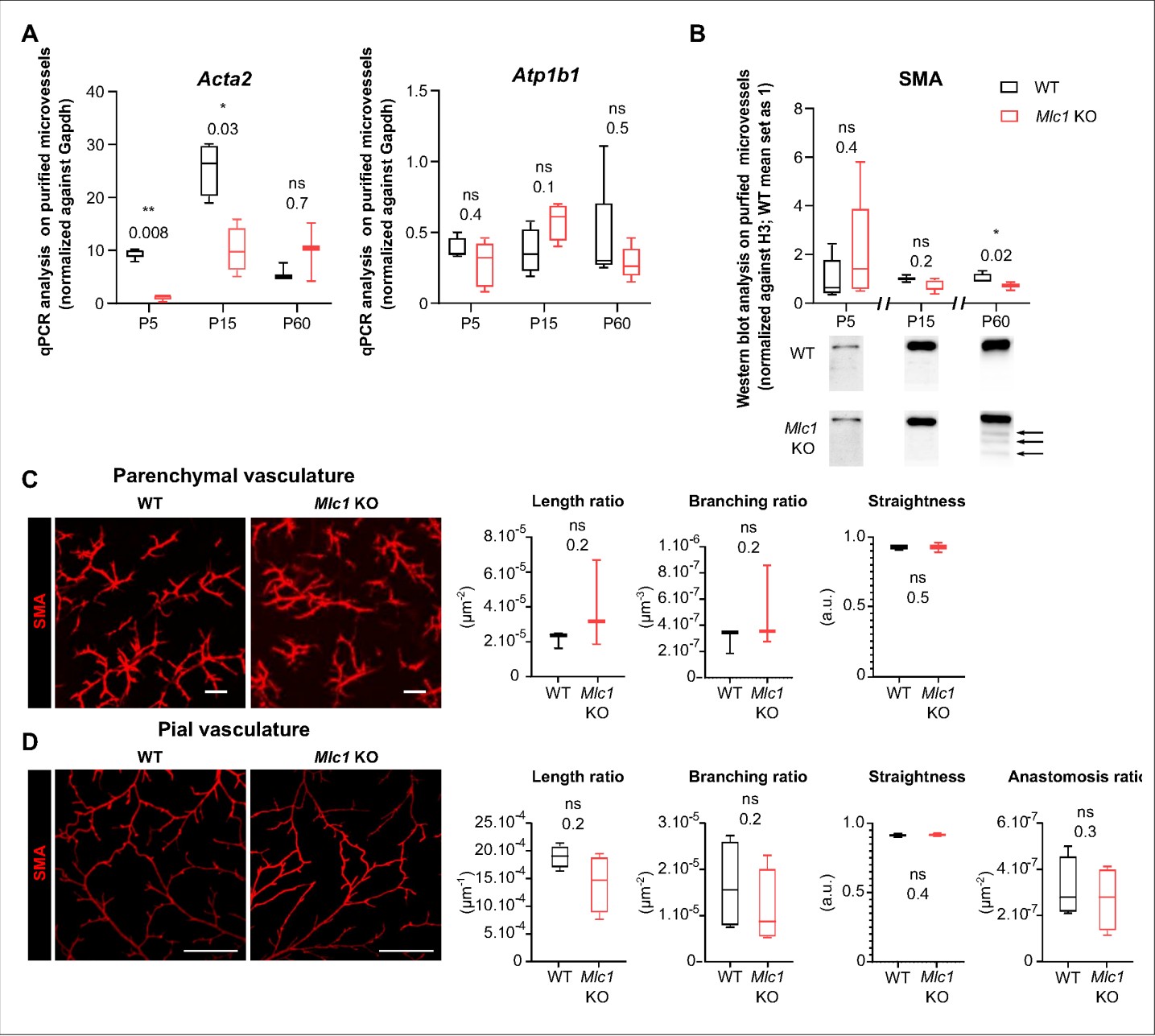

**Figure 2.** MLC1 is crucial for the molecular maturation of vascular smooth muscle cell contractility. (**A**) qPCR results for Acta2 (encoding SMA) and Atp1b1 in microvessels purified from WT and Mlc1 KO whole brains on P5, P15, and p60. Signals are normalized against Gapdh. Two-tailed Mann-Whitney test. The data are represented in a Tukey box plot (n = 3 to 5 samples per genotype; number of brains pooled per sample: 5 for P5; 3 for P15; 2 for P60). (**B**) Western blot detection and analysis of SMA in protein extracts from microvessels purified from WT and Mlc1 KO whole brains on P5, P15, and P60. Arrows indicate abnormally low molecular weight SMA-positive bands. Signals were normalized against histone H3. Two-tailed Mann-Whitney test. The data are represented in a Tukey box plot (n = 4 or 5 samples per genotype; number of brains pooled per sample: 5 for P5; 3 for P15; 2 for P60). (**C.D**) Representative 3D images of the vascular smooth muscle cell arterial network in cleared somatosensory cortex. Parenchymal (Z stack 600 μm; scale bar: 100 μm) (**C**) and pial vessels (Z stack 50 μm; scale bar: 500 μm) (**D**) samples in 2-month-old WT and Mlc1 KO mice after immunolabeling for SMA. Comparative analysis of arterial length, branching, tortuosity, and anastomosis in WT mice (black boxes) and Mlc1 KO mice (red boxes) in the cortical parenchyma (**C**), normalized on sample volume, and at the cortical surface (**D**), normalized on sample surface. One-tailed Mann-Whitney test. The data are represented in a Tukey box plot (n = 3 mice per genotype). The data are given in *Figure 2—source data 1* *, p ≤ 0.05, **, p ≤ 0.01, ***, p ≤ 0.001, and ns: not significant.

The online version of this article includes the following figure supplement(s) for figure 2:

**Source data 1.** MLC1 is crucial for the molecular maturation of vascular smooth muscle cell contractility.

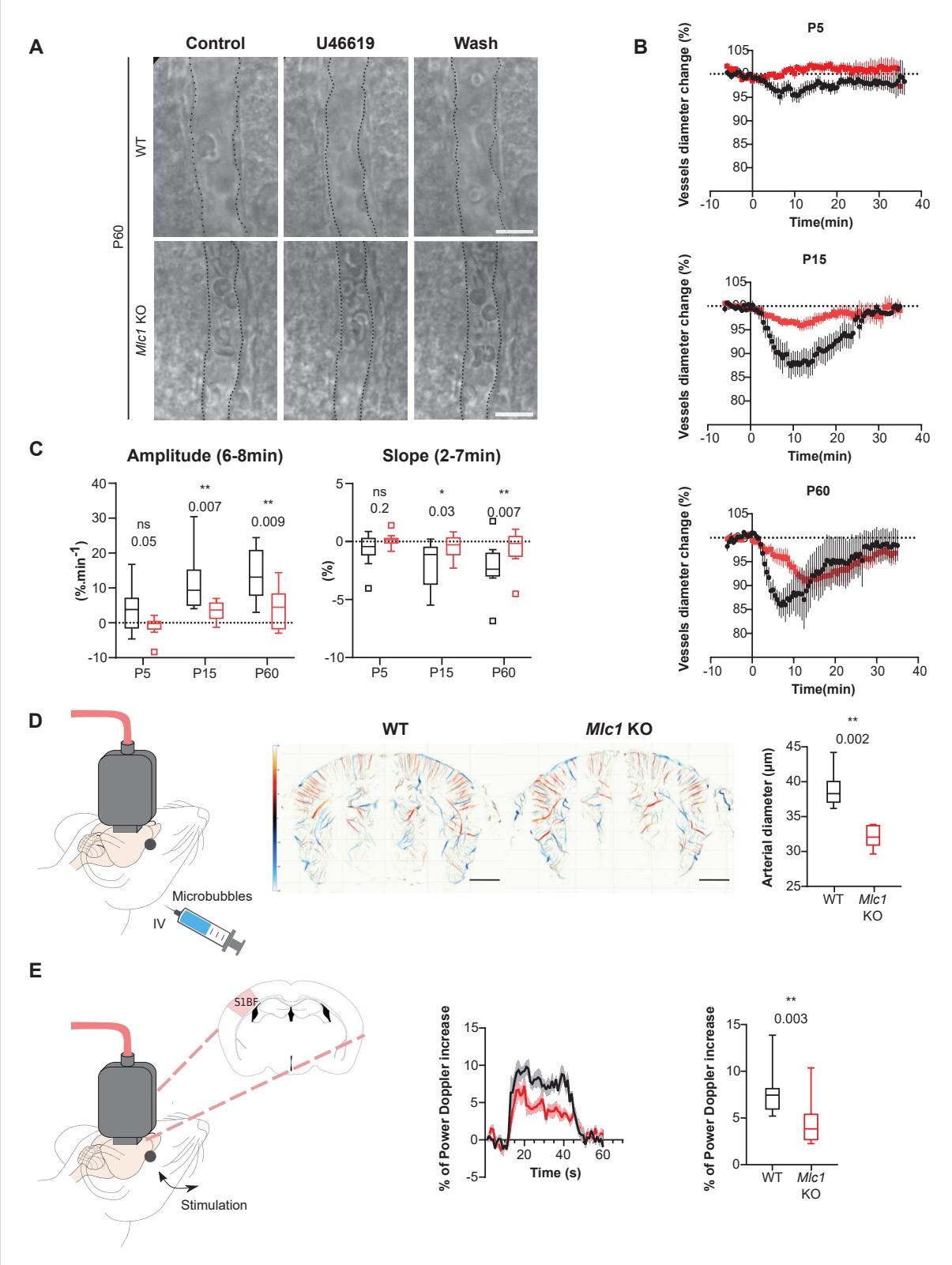

**Figure 3.** MLC1 is crucial for the postnatal acquisition of vascular smooth muscle cell contractility, arterial diameter, and neurovascular coupling. (**A–C**) Ex vivo analysis of mean vascular constriction and dilation upon application of U46619 (50 nM) to somatosensory cortical slices from postnatal day (P)5, P15, and P60 WT mice (black traces) and *Mlc1* KO mice (red traces). (**A**) Representative infrared images of a cortical penetrating arteriole constriction in response to bath application of U46619 and dilation upon washing at P60. The vessel lumen is indicated by dotted lines. Scale bar: 10 μm. (**B**)

*Figure 3 continued on next page*

*Figure 3 continued*

Contraction and dilation slopes on P5, P15, and P60. 0 min corresponds to the addition of U46619 to the recording chamber medium. The data are presented as the mean ± SEM. (**C**) Analysis of the amplitude and slope of the contraction. Two-tailed Mann–Whitney test. The data are represented in a Tukey box plot (n = 13 vessels from WT and 13 *Mlc1* KO mice on P5; n = 9 vessels from WT and 12 *Mlc1* KO mice at P15; n = 8 vessels from WT and 11 *Mlc1* KO mice at P60; 3 mice per group). (**D**) In vivo ultrasound localization microscopy measurement of cortical arterial vessel diameter after intravenous microbubble injection in 2-month-old WT and *Mlc1* KO mice. Left: schematic representation of the experiment; middle: cerebrovascular maps of WT and *Mlc1* KO mice, showing the arterial (in red) and venous (in blue) velocities in mm/s (scale bar: 0.15 cm); right: measurement of the penetrating arteries' diameter using ultrasound localization microscopy imaging of an injected microbubbles. The data are represented in a Tukey box plot. Two-tailed Mann–Whitney test (n = 6 WT mice and *Mlc1* KO mice each). (**E**) In vivo functional ultrasound analysis of cerebral blood flow in the somatosensory cortex after whisker stimulation. Left: schematic representation of the experiment (S1bf: bregma –1.5 mm, somatosensory barrel field cortex); middle: functional ultrasound power Doppler signal traces during whisker stimulation of 2-month-old WT mice (in black) and *Mlc1* KO mice (in red) mice. Baselines before stimulation are aligned; Right: quantification of the normalized percentage of cerebral blood flow variation following whisker stimulation. Two-tailed Mann–Whitney test. The gray areas around the curves correspond to the SEM (n = 11 WT mice and 12 *Mlc1* KO mice). The data are given in *Figure 3—source data 1*. *p≤0.05, **p≤0.01, ***p≤0.001, and ns: not significant. Data presented in (**A–C**) are also included in *Slaoui et al., 2021*. These panels are not available under the terms of a Creative Commons Attribution License, and further reproduction of these images requires permission from the copyright holder.

The online version of this article includes the following figure supplement(s) for figure 3:

**Source data 1.** MLC1 is crucial for the postnatal acquisition of vascular smooth muscle cell contractility, arterial diameter, and neurovascular coupling.

neurovascular coupling, that is, variations in cerebral blood flow in the barrel cortex in response to whisker stimulation in 2-month-old WT and *Mlc1* KO mice (*Bertolo et al., 2021*; *Hingot et al., 2020*; *Osmanski et al., 2014*; *Figure 3E*, *Figure 3—source data 1*). The increase in cerebral blood flow evoked by whisker stimulation was significantly smaller in *Mlc1* KO mice than in WT mice, indicating that neurovascular coupling was impaired in the KO mice (*Figure 3E*, *Figure 3—source data 1*).

In conclusion, the absence of MLC1 impairs the postnatal acquisition of contractile properties by vascular smooth muscle cells and impedes blood perfusion, vessel diameter, and neurovascular coupling.

## The absence of MLC1 alters the perivascular astrocytic processes' molecular maturation and organization

We next characterized the perivascular astrocytic processes in *Mlc1* KO mice by focusing on membrane proteins known to be strongly expressed in these structures (*Cohen-Salmon et al., 2021*): the water channel aquaporin 4 (Aqp4), the gap junction protein connexin 43 (Cx43), the adhesion protein GlialCAM, and the inward rectifier potassium channel Kir4.1. We first used immunofluorescence to analyze the proteins' location in perivascular astrocytic processes on brain sections on P5, P15, and P60; Kir4.1 was analyzed from P15 onward since it is barely detectable before this stage (*Figure 4A*). The vessels were counterstained with isolectin B4. GlialCAM (whose location at the astrocyte endfoot membrane and immunolabeling depends on MLC1) was not detected around vessel in *Mlc1* KO mice at any stage, as described previously (*Hoegg-Beiler et al., 2014*). Interestingly, perivascular Aqp4 and Cx43 were almost undetectable on P5 in *Mlc1* KO mice but were detected on perivascular astrocytic membranes from P15 onward (*Figure 4A*). Kir4.1 was present on perivascular astrocytic membranes in *Mlc1* KO mice, although the immunofluorescence was less intense. To further quantify these results, we first used western blots to assess the levels of Aqp4, Cx43, GlialCAM, and Kir4.1 in whole-brain protein extracts (*Figure 4B*, *Figure 4—source data 1*). In *Mlc1* KO mice, we observed lower levels of Aqp4, Cx43, and GlialCAM on P5 only and lower levels of Kir4.1 on P15 and P60 (*Figure 4B*). Using stimulated emission depletion (STED) and confocal microscopy, we next quantified the perivascular location of Aqp4, Cx43, GlialCAM, and Kir4.1 in WT and *Mlc1* KO mice on P60 (*Figure 4C*). The vessel's surface was stained by isolectin B4. The Aqp4 signal was similar in the WT and *Mlc1* KO samples. In the *Mlc1* KO mice, however, the perivascular Cx43 puncta were larger and denser (*Figure 4C*, *Figure 4—source data 1*). GlialCAM labeling was almost undetectable in *Mlc1* KO mice. Kir4.1 perivascular puncta were the same size in both the WT and *Mlc1* KO samples but were less dense in *Mlc1* KO mice.

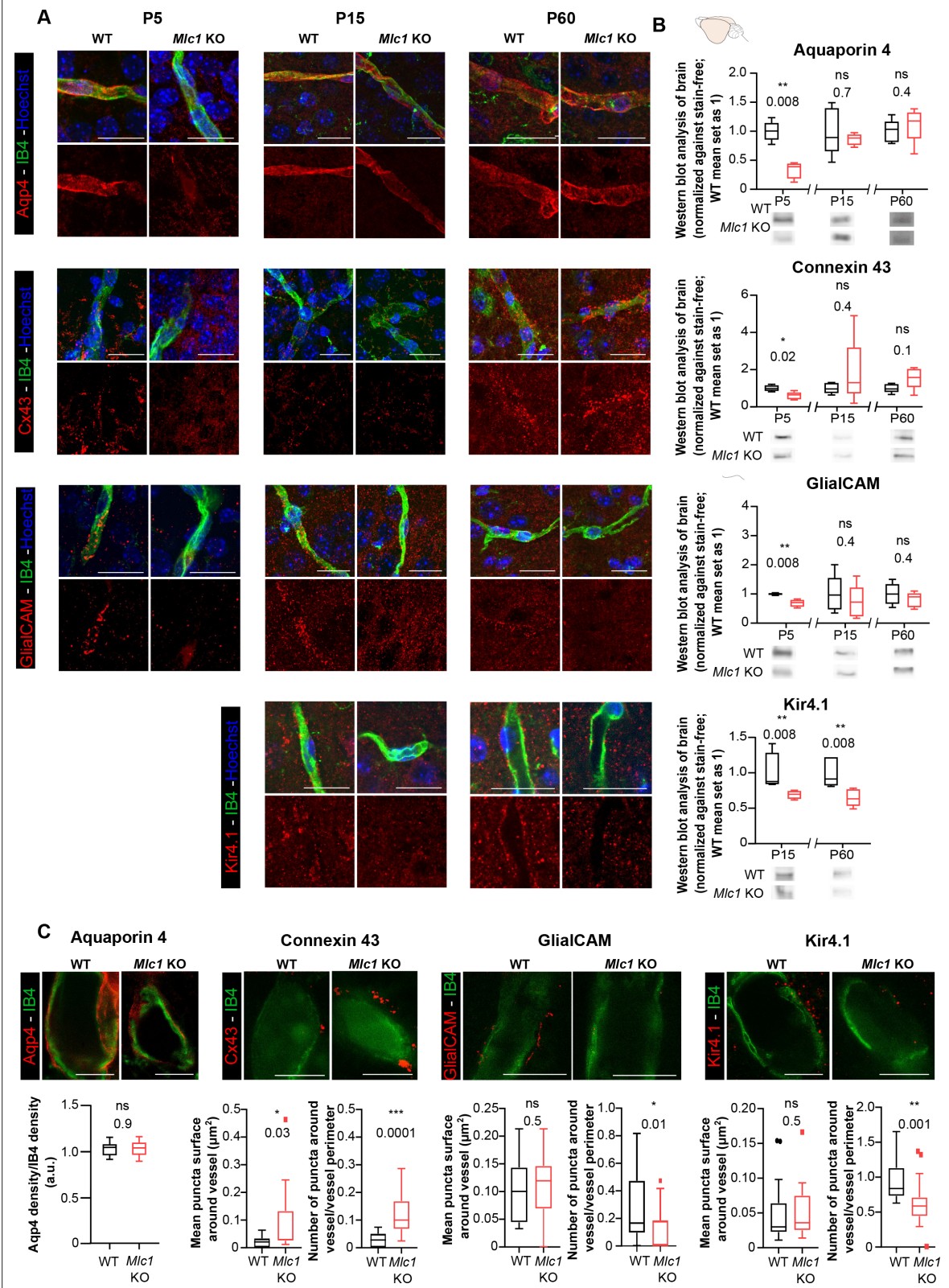

**Figure 4.** The absence of MLC1 alters the molecular maturation of the perivascular astrocytic processes. (**A**) Representative confocal projection images of the immunofluorescent detection of Aqp4, Cx43, GlialCAM, and Kir4.1 (in red) on brain cortex sections from WT and *Mlc1* KO mice on postnatal day (P)5 (except for Kir4.1), P15, and P60. Vessels were stained with isolectin B4 (IB4) (in green), and nuclei were stained with Hoechst dye (in blue). Scale bar: 20 μm. (**B**) Western blot detection and analysis of Aqp4, Cx43, GlialCAM, and Kir4.1 in whole-brain protein extracts from WT and *Mlc1* KO mice

*Figure 4 continued on next page*

*Figure 4 continued*

on P5 (except for Kir4.1), P15, and P60. Two-tailed Mann–Whitney test. The data are represented in a Tukey box plot (for whole brain: n = 5 samples per genotype [one mouse per sample]). (**C**) Representative stimulated emission depletion (STED) images and quantification of the immunofluorescent detection of Aqp4, Cx43, GlialCAM, and Kir4.1 (in red) on brain cortex sections from WT and *Mlc1* KO mice on P60. Vessels were stained with IB4 (in green). Intravascular diffusion of the IB4 fluorescence is an artifact of STED. Scale bar: 5 μm. Two-tailed Mann–Whitney test. The data are represented in a Tukey box plot (aquaporin 4: n = 18 *Mlc1* KO vessels; n = 18 WT vessels; n = 3 mice per genotype; connexin 43: n = 15 *Mlc1* KO vessels; n = 15 WT vessels; n = 3 mice per genotype; GlialCAM: n = 14 *Mlc1* KO vessels; n = 14 WT vessels; n = 3 mice per genotype; Kir4.1: n = 18 *Mlc1* KO vessels; n = 17 WT vessels; n = 3 mice per genotype). The data are given in *Figure 4—source data 1*. *p≤0.05, **p≤0.01, ***p≤0.001, and ns: not significant.

The online version of this article includes the following figure supplement(s) for figure 4:

**Source data 1.** The absence of MLC1 alters the molecular maturation of the perivascular astrocytic processes.

These results show that the absence of MLC1 is associated with lower expression of Kir4.1, the delayed expression of Aqp4 and Cx43, the disruption (from P5 onward) of the GlialCAM's membrane anchoring, and impairment of Cx43's perivascular organization. The data suggest that MLC1 is required for the development and maintenance of the perivascular astrocytic processes' molecular organization.

## The absence of MLC1 alters the perivascular astrocytic processes' mechanical cohesiveness

We had demonstrated previously that perivascular astrocytic processes and the associated neuronal fibers remained attached to vessels during their mechanical purification (*Boulay et al., 2015*; *Figure 5E*). The level of Aqp4 in the whole-brain extract on P60 was similar in WT and *Mlc1* KO mice (*Figure 4B*). However, the level of Aqp4 in the extract of mechanically purified brain microvessels was significantly lower in *Mlc1* KO mice than in WT mice (*Figure 5A*, *Figure 5—source data 1*). The same was true for neurofilament protein M (NF-M), a neuronal-specific intermediate filament protein present in neuronal fibers abutting perivascular astrocytic processes. NF-M was similarly present in whole-brain extracts from WT and *Mlc1* KO mice but was significantly less present in P60 *Mlc1* KO microvessels (*Figure 5B*, *Figure 5—source data 1*). These data suggested that the perivascular astrocytic processes and the associated neuronal fibers had detached from *Mlc1* KO brain vessels when the latter were mechanically purified. To further validate these results, we assessed the levels of Aqp4 and NF-M immunofluorescence in mechanically purified microvessels from P60 *Mlc1* KO and WT mice (*Figure 5C and D*, *Figure 5—source data 1*). The vessels were counterstained with isolectin B4. In the *Mlc1* KO sample, Aqp4 and NF-M immunolabeling present at the surface of the purified vessels was discontinuous and less intense than in the WT sample (*Figure 5C and D*, *Figure 5—source data 1*).

Taken as a whole, these results suggest that MLC1 influences the mechanical cohesiveness of perivascular astrocytic processes at the vessel surface (*Figure 5E*).

## The absence of MLC1 alters the development and maintenance of astrocyte morphology and polarity

Since tissue cohesivity, cell morphology, and polarity are interdependent, we hypothesized that the absence of MLC1 could perturb the astrocytes' morphology and polarity. We addressed this question by performing a Sholl analysis of glial fibrillary acid protein (GFAP)-immunolabeled astrocytic ramifications in the CA1 region of the hippocampus on P60 (*Figure 6A*). *Mlc1* KO astrocytes displayed a greater number of processes located between 15 and 25 μm from the soma (*Figure 6A*, *Figure 6—source data 1*). In the hippocampal stratum radiatum, GFAP-positive astrocytic processes are normally polarized perpendicular to the pyramidal cell layer (*Nixdorf-Bergweiler et al., 1994*). We evaluated this preferential orientation and measured the polarity index, which corresponds to the ratio between parallel (axial) and perpendicular (lateral) crossing points between GFAP-positive processes and a grid oriented with the pyramidal cell layer (*Figure 6C*). We found that both *Mlc1* KO and WT astrocytes were equally well oriented, with a polarity index >1 (*Figure 6C*, *Figure 6—source data 1*). However, when taking hippocampal vessels as the reference, *Mlc1* KO astrocytes were abnormally oriented toward vessels in the axial plane (*Figure 6B*, *Figure 6—source data 1*).

MLC1 expression is progressive; in the mouse, it starts at P5 and finishes at P15 (*Gilbert et al., 2019*). Astrocyte ramification in the stratum radiatum increases greatly between P8 and P16

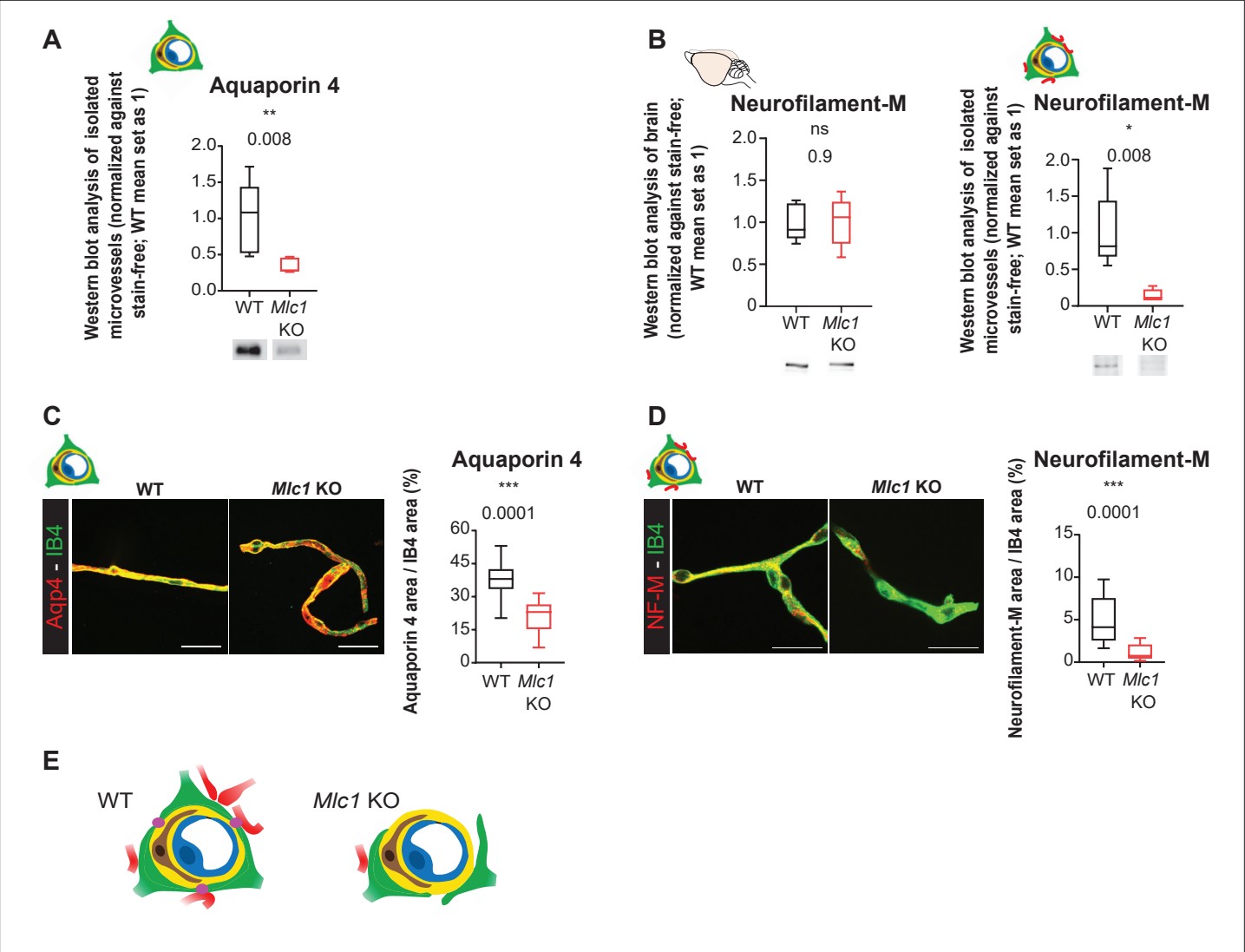

**Figure 5.** The absence of MLC1 alters the perivascular cohesiveness of astrocytic processes. (**A, B**) Western blot detection and analysis of Aqp4 (**A**) and NF-M (**B**) in protein extracts from microvessels purified from WT and *Mlc1* KO whole brains on postnatal day (P)60 (see *Figure 4C* for Aqp4 detection in whole brain on P60). The signals were normalized against stain-free membranes. Two-tailed Mann–Whitney test. The data are represented in a Tukey box plot (n = 5 samples per genotype [ two mice per microvessels sample]). (**C, D**) Representative confocal projection images of the immunofluorescent detection and quantification of Aqp4 (**C**) and NF-M (**D**) (in red) on microvessels purified from WT and *Mlc1* KO whole brain on P60. Vessels were stained with isolectin B4 (IB4) (in green). Scale bar: 20 µm. Two-tailed Mann–Whitney test. The data are represented in a Tukey box plot (aquaporin 4: n = 17 *Mlc1* KO images; n = 22 WT images; n = 3 mice per genotype; NF-M: n = 15 *Mlc1* KO images; n = 16 WT images; n = 3 mice per genotype; each image shows between 1 and 4 vessels). The data are given in *Figure 5—source data 1*. *p≤0.05, **p≤0.01, ***p≤0.001, and ns: not significant. (**E**) Schematic interpretation of the data. In *Mlc1* KO mice, the perivascular astrocytic processes (in green) and neuronal associated fibers (in red) are lost during the microvessel purification process (basal lamina in yellow, mural cells in brown, EC in blue). MLC1 is represented by pink dots in the WT.

The online version of this article includes the following figure supplement(s) for figure 5:

**Source data 1.** The absence of MLC1 alters the perivascular cohesiveness of astrocytic processes.

(*Nixdorf-Bergweiler et al., 1994*). We therefore wondered whether the abnormal morphology and polarity observed in adult *Mlc1* KO astrocytes might be caused by a developmental defect. Hence, we analyzed the morphology and orientation of GFAP-immunolabeled astrocytic processes on P10 and P15. As previously observed on P60, *Mlc1* KO astrocytes had a larger number of processes located 10–25 µm from the soma (*Figure 6D and F*, *Figure 6—source data 1*, *Figure 6—figure supplement 1*). On P10 and P15, the *Mlc1* KO processes were also abnormally oriented toward the hippocampal

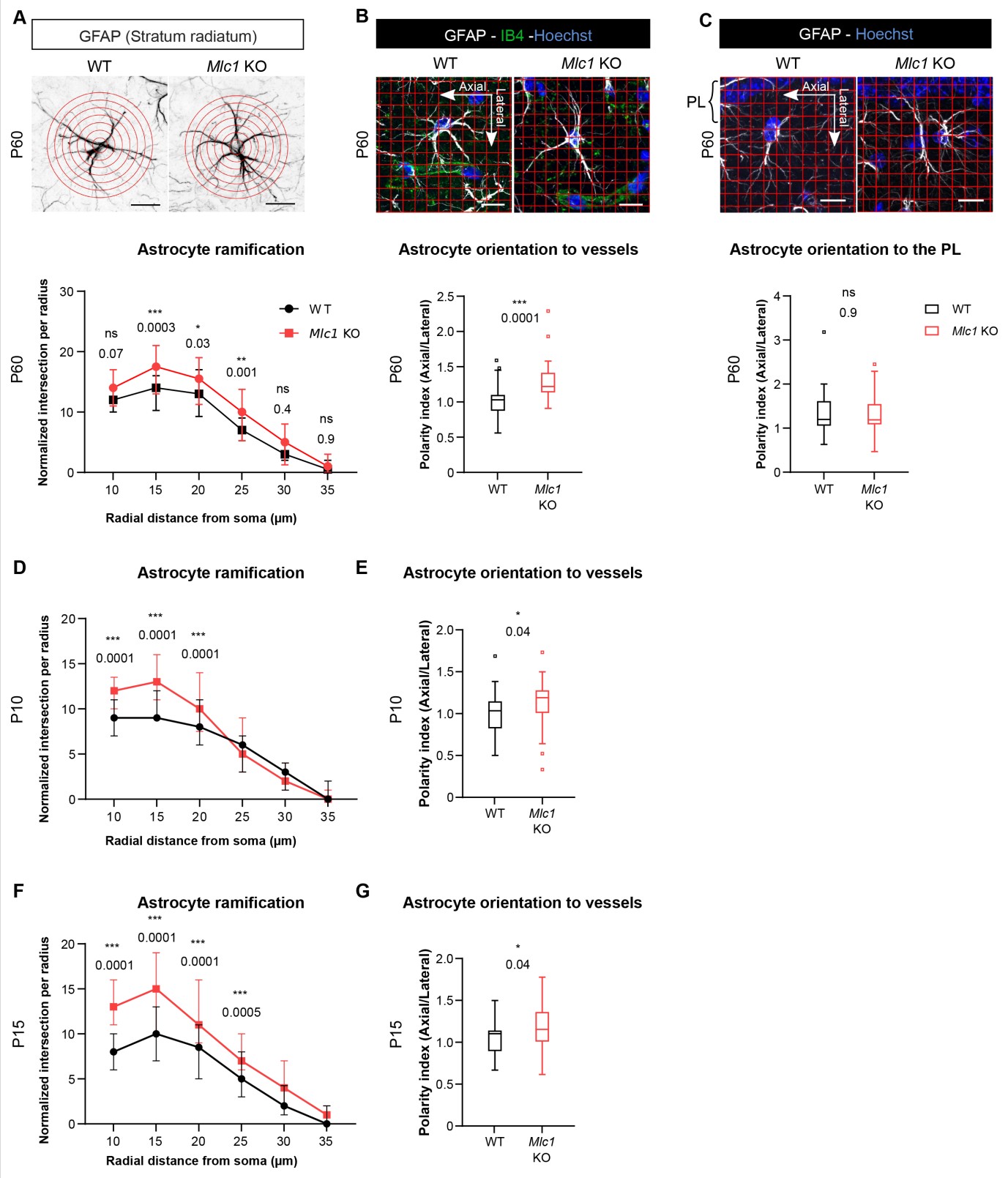

**Figure 6.** The absence of MLC1 alters the postnatal development and maintenance of astrocyte morphology and polarity. (**A**) A Sholl analysis of the ramification of hippocampal CA1 astrocytes immunolabeled for GFAP (in black) in WT and *Mlc1* KO postnatal day (P)60 mice. The concentric circles start from the astrocyte's soma. Scale bar: 20 µm. A two-way analysis of variance, followed by a Bonferroni *post hoc* test. The data are presented as the median± quartiles. (n = 48 *Mlc1* KO cells; n = 44 WT cells; n = 3 mice per genotype). (**B**) Grid baseline analysis of the orientation of the GFAP-

*Figure 6 continued on next page*

*Figure 6 continued*

immunolabeled astrocytic processes (in white) toward vessels labeled with isolectin B4 (in green) in WT and *Mlc1* KO P60 mice. The nuclei were labeled with Hoechst dye (in blue). Scale bar: 20 µm. The polarity index is the ratio between axial GFAP contacts and lateral GFAP contacts. A polarity index of 1 means that there is no polarity. Two-tailed Mann–Whitney test. The data are represented in a Tukey box plot (n = 41 *Mlc1* KO cells; n = 41 WT cells; 3 mice per genotype). (**C**) Grid baseline analysis of the orientation of the GFAP-immunolabeled astrocytic processes (in white) toward the hippocampal pyramidal cell layer in WT and *Mlc1* KO P60 mice. The nuclei were labeled with Hoechst dye (in blue). Scale bar: 20 µm. The polarity index is the ratio between axial GFAP contacts and lateral GFAP contacts. A polarity index of 1 means that there is no polarity. Two-tailed Mann–Whitney test. The data are represented in a Tukey box plot (n = 52 *Mlc1* KO cells; n = 47 WT cells; 3 mice per genotype). (**D–F**) Quantitative analyses of astrocyte ramification on P10 (**D**) and P15 (**F**). A two-way analysis of variance, followed by a Bonferroni *post hoc* test (P10: n = 53 *Mlc1* KO cells; n = 49 WT cells; n = 3 mice per genotype; P15: n = 51 *Mlc1* KO cells; n = 50 WT cells; n = 3 mice per genotype). See *Figure 6—figure supplement 1* for representative images. (**E–G**) Quantitative analysis of astrocyte process orientation toward vessels on P10 (**E**) and P15 (**G**). See *Figure 6—figure supplement 1* for representative images. The polarity index is the ratio between axial GFAP contacts and lateral GFAP contacts. A polarity index of 1 means that there is no polarity. Two-tailed Mann–Whitney test. The data are represented in a Tukey box plot (P10: n = 35 *Mlc1* KO cells; n = 26 WT cells; n = 3 mice per genotype; P15: n = 48 *Mlc1* KO cells; n = 46 WT cells; n = 3 mice per genotype). The data are given in *Figure 6—source data 1*. *p≤0.05, **p≤0.01, ***p≤0.001, and ns: not significant.

The online version of this article includes the following figure supplement(s) for figure 6:

**Source data 1.** The absence of MLC1 alters the postnatal development and maintenance of astrocyte morphology and polarity.

**Figure supplement 1.** The absence of MLC1 alters the postnatal development of astrocyte morphology and polarity.

vessels, relative to WT astrocytes (*Figure 6E and G*, *Figure 6—source data 1*, *Figure 6—figure supplement 1*).

Taken as a whole, these results indicate that MLC1 is required for the development and maintenance of astrocyte morphology and polarity.

## The absence of MLC1 impacts the gliovascular unit's organization and development

We next used transmission electron microscopy to analyze the ultrastructural morphology of perivascular astrocytic processes in the cortex and hippocampus of 2-month-old WT and *Mlc1* KO mice, with a focus on vessels up to 10 µm in diameter (*Figure 7*). In WT mice, endothelial cells were joined by tight junctions and were totally covered by perivascular astrocytic processes, which themselves were joined by gap junctions. Astrocytes and endothelial cells were separated by a thin, homogeneous, regular basal lamina (*Figure 7A*). In *Mlc1* KO mice, the endothelium appeared to be unaltered: normal tight junctions and basal lamina, and no accumulation of intracellular vesicles (*Figure 7B–E*). However, the astrocytes' perivascular organization was drastically modified (*Figure 7B–E*). We observed perivascular astrocytic processes surrounded by basal lamina (*Figure 7C*, *Figure 7—figure supplement 1*) or stacked on top of each other and joined by gigantic gap junction plaques (*Figure 7D*). Some perivascular astrocytic processes interpenetrated each other (*Figure 7—figure supplement 1*). No swelling was observed in *Mlc1* KO perivascular astrocytic processes (*Figure 7B–E and G*, *Figure 7—source data 1*) (edematous perivascular astrocytic processes were observed in 1-year-old *Mlc1* KO mice; *Figure 7—figure supplement 2*, *Figure 7—figure supplement 2—source data 1*). Strikingly, astrocyte coverage in *Mlc1* KO mice was often discontinuous. In the free spaces, axons (recognizable by their microtubules) (*Figure 7B and E*) and synapses (recognizable by the large number of vesicles in the presynaptic part and their electron dense postsynaptic density (*Figure 7—figure supplement 1*)) were found to be in direct contact with the endothelial basal lamina. Accordingly, the percentage of microvessels in which the endothelial basal lamina was in direct contact with at least one neuronal process was higher in *Mlc1* KO mice than in WT mice (*Figure 7F*).

The time course of perivascular astrocyte coverage has not previously been described. Here, we used transmission electron microscopy to quantify the percentage of the perivascular diameter covered by perivascular astrocytic processes in *Mlc1* KO and WT cortex and hippocampus on P5, P10, P15, and P60 (*Figure 7H*, *Figure 7—figure supplement 1*, *Figure 7—source data 1*). On P5, the perivascular astrocytic processes covered about half of the vessel's circumference, and the WT and *Mlc1* KO samples did not differ significantly in this respect (*Figure 7H*). The perivascular astrocyte coverage increased dramatically between P5 and P15 and was almost complete on P15 in WT mice (*Figure 7H*). In contrast, *Mlc1* KO mice displayed a lower percentage of perivascular astrocytic process coverage

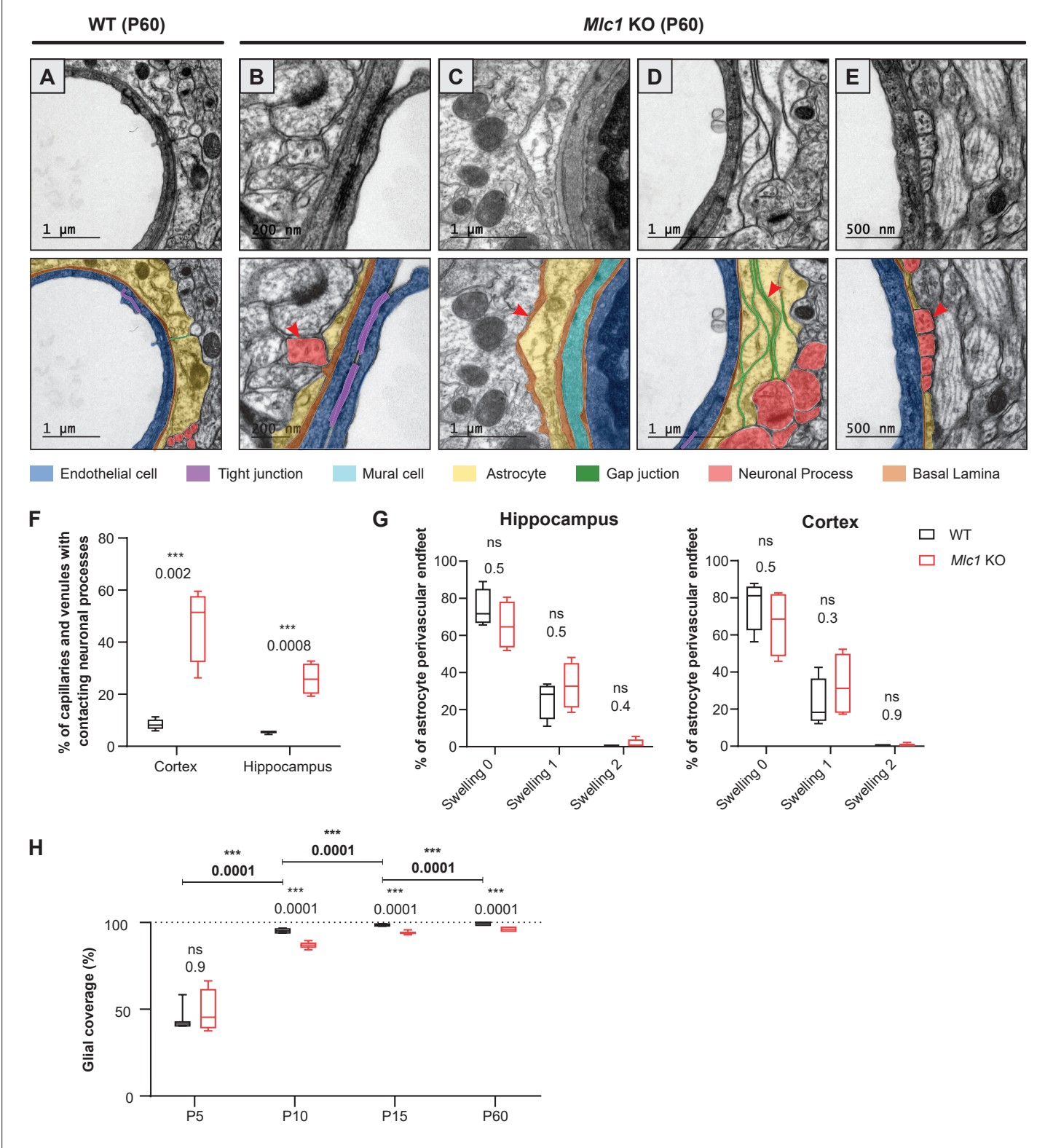

**Figure 7.** The absence of MLC1 impacts the organization and development of the gliovascular unit. (**A–E**) Representative transmission electron microscopy images of the gliovascular unit in the hippocampus of postnatal day (P)60 WT and *Mlc1* KO mice (n = 3 mice per genotype). Images are presented in pairs, with artificial colors in the lower panel: perivascular astrocytic processes in yellow, gap junctions in green, axons or synapses in red, mural cells in light blue, endothelial cells in dark blue, the basal lamina in brown, and tight junctions in purple. (**A**) In WT mice, perivascular astrocytic

*Figure 7 continued on next page*

*Figure 7 continued*

processes fully cover endothelial cells linked by a tight junction and surrounded by a continuous basal lamina. (**B–E**) Data from *Mlc1* KO mice. (**B**) Perivascular astrocytic processes are separated by an axon, which contacts the endothelial basal lamina (red arrowhead). (**C**) A perivascular astrocytic process is surrounded by the basal lamina (red arrowhead). (**D**) Several perivascular astrocytic processes are stacked on top of each other and are linked by extended gap junctions (red arrowhead). (**E**) Perivascular astrocytic processes are separated by four axons (red arrowhead), which are in direct contact with the vascular basal lamina. (**F**) Quantification of capillaries and venules contacted by neural processes (axons or synapses) in P60 mice. Two-tailed Student's t-test. The data are represented in a Tukey box plot (n = 399 *Mlc1* KO cortical vessels; n = 301 *Mlc1* KO hippocampal vessels; n = 286 WT cortical vessels; n = 287 WT hippocampal vessels; n = 3 mice per genotype). (**G**) Percentage of vessels contacted by a normal perivascular astrocytic process (swelling = 0), a moderately swollen perivascular astrocytic process (swelling = 1), or an edematous perivascular astrocytic process (swelling = 2) in the hippocampus and cortex of P60 mice. Two-tailed Mann–Whitney test. The data are represented in a Tukey box plot (n = 399 *Mlc1* KO cortical vessels; n = 301 *Mlc1* KO hippocampal vessels; n = 286 WT cortical vessels; n = 287 WT hippocampal vessels; n = 3 mice per genotype). (**H**) Percentage of the vessel diameter covered by perivascular astrocytic processes in the cortex of WT and *Mlc*1 KO mice on P5, P10, P15, and P60. Two-tailed Mann–Whitney test. The data are represented in a Tukey box plot (n = 46 vessels from WT mice and 68 *Mlc1* KO mice on P5, n = 3 mice per genotype; n = 121 vessels from WT mice and 81 *Mlc1* KO mice on P10, n = 3 mice per genotype; n = 207 vessels from WT mice and 144 *Mlc1* KO mice at P15, n = 4 mice per genotype; n = 143 vessels from WT mice and 134 *Mlc1* KO mice at P60, n = 3 mice per genotype). Representative transmission electron microscopy images of the gliovascular interface in the cortex of WT and *Mlc1* KO mice on P10 and P15 are presented in *Figure 7—figure supplement 1*. The data are given in *Figure 7—source data 1*. *p≤0.05, **p≤0.01, ***p≤0.001, and ns: not significant.

The online version of this article includes the following figure supplement(s) for figure 7:

**Source data 1.** The absence of MLC1 impacts the organization and development of the gliovascular unit.

**Figure supplement 1.** Examples of changes in the architecture of the gliovascular unit in *Mlc1* KO mice.

**Figure supplement 2.** Astrocytic perivascular endfeet are swollen in 1-year-old *Mlc1* KO mice.

**Figure supplement 2—source data 1.** Astrocytic perivascular endfeet are swollen in 1-year-old *Mlc1* KO mice.

from P10 onward, and a large number of neuronal processes were inserted into the noncovered areas of the vessels (*Figure 7H*, *Figure 7—figure supplement 1*, *Figure 7—source data 1*).

Our results demonstrate for the first time that the perivascular astrocyte coverage increases rapidly between P5 and P15. This process is impaired in *Mlc1* KO mice and results in incomplete perivascular astrocytic process coverage and direct contact between infiltrating neuronal processes and the endothelial basal lamina (the processes normally remain behind the perivascular astrocyte layer). Taken as a whole, these results indicate that (i) the absence of MLC1 in the gliovascular unit greatly alters the perivascular astrocytic processes' morphology and coverage and (ii) MLC1 is critical for the normal postnatal development of perivascular astrocyte coverage.

## The absence of MLC1 modifies the parenchymal circulation of cerebrospinal fluid

Several groups have reported a causal link between perivascular astrocytic process disorganization and impaired parenchymal cerebrospinal fluid (CSF) transport (*Haj-Yasein et al., 2012*; *Kress et al., 2014*). We tested this hypothesis by measuring the parenchymal distribution of DOTA-gadolinium (DOTA-Gd) injected in CSF through the cisterna magna in 2-month-old mice using T1-weighted magnetic resonance imaging (*Figure 8A*). Contrast-enhanced T1 mapping was used to quantify changes of the concentration of DOTA-Gd within the brain tissue over time. In line with the conventional models of cerebrospinal solute circulation, tracers injected into the cisterna magna dispersed into the subarachnoid space and then entered the parenchyma through the perivascular spaces (*Figure 7B*; *Iliff et al., 2012*; *Iliff et al., 2013*). As expected, a high concentration of DOTA-Gd was detected at the border of the brain as early as 10 min after injection in both WT and *Mlc1* KO mice; this reflected the initial dispersion of contrast within the subarachnoid space. 45 min after injection, the DOTA-Gd concentration in brain tissue rose as it penetrated into the parenchyma through the perivascular spaces (*Figure 8C*). The dispersion kinetics for each route depends on anatomic differences between regions of the brain, such as the presence or absence of a perivascular space and the topology of the vascular network (*Iliff et al., 2012*; *Iliff et al., 2013*). For both the WT and *Mlc1* KO genotypes, the distribution of DOTA-Gd appeared to be in line with the literature data (*Figure 8C*). However, examination of the quantitative maps suggested that DOTA-Gd transport into the CSF was less intense in *Mlc1* KO mice than in WT mice. We also analyzed the time courses of DOTA-Gd dispersion in the cerebellum, midbrain, septal area, and cortex (*Figure 8D–G*, *Figure 8—source data 1*). Relative to WT mice, the mean DOTA-Gd concentration in *Mlc1* KO mice was lower in the midbrain (*Figure 8E*, *Figure 8—source data 1*) and

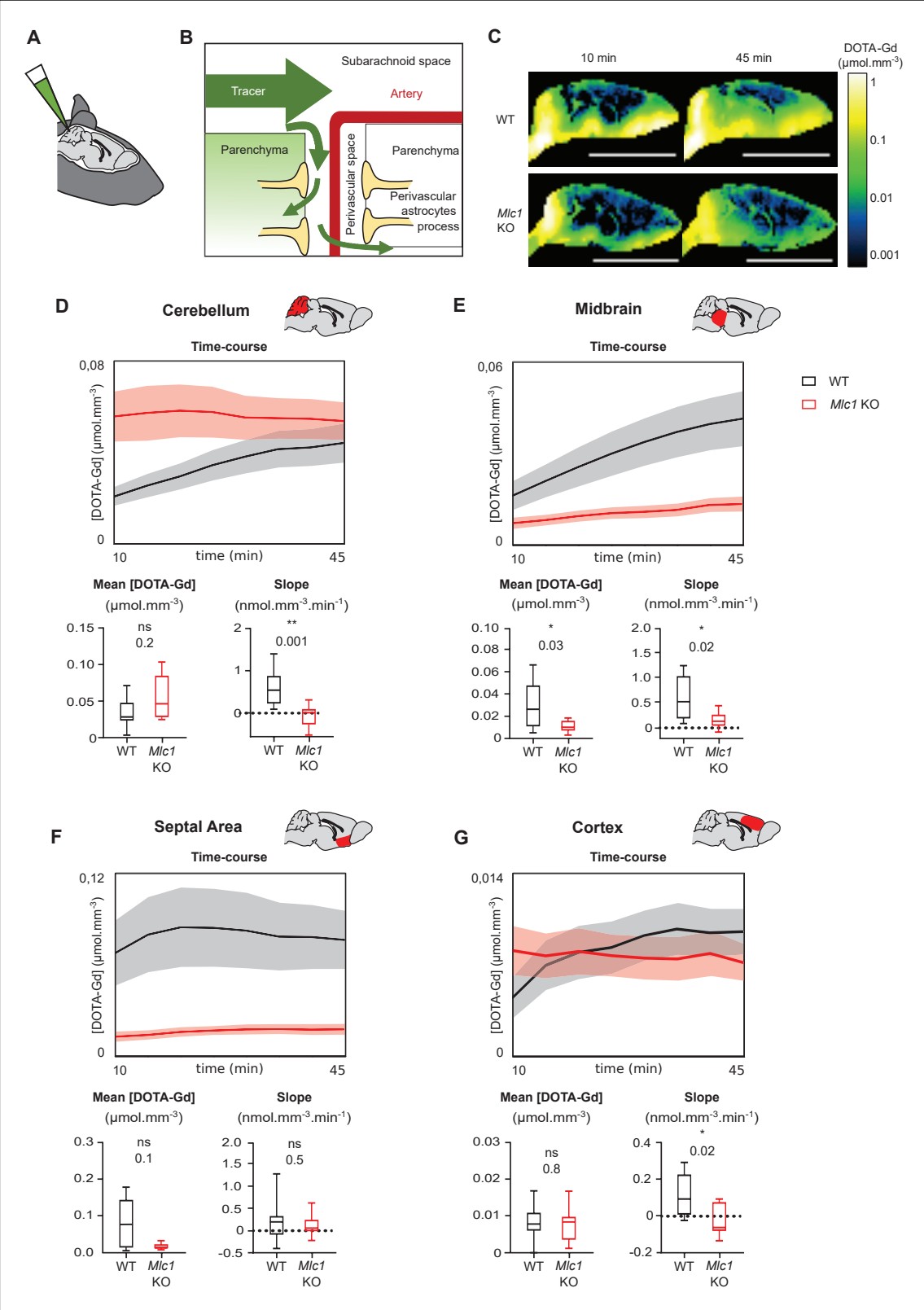

**Figure 8.** Contrast-enhanced magnetic resonance imaging reveals a low level of tracer dispersion from the cerebrospinal fluid (CSF) into the parenchyma in *Mlc1* KO mice. (**A, B**) Schematic representation of injecting 1 μL DOTA-Gd into the mouse's CSF through the cisterna magna (**A**) and the disperse of the tracer through the brain into the subarachnoid space before entering the deep parenchyma through the perivascular spaces (**B**). (**C**) Quantitative contrast maps in WT and *Mlc1* KO mice, 10 and 45 min after contrast injection (scale bar: 1 cm). (**D–G**) Based on the dynamic acquisitions,

*Figure 8 continued on next page*

*Figure 8 continued*

the changes over time in contrast agent concentration were extracted, and the mean contrast concentration and the contrast slope were calculated for the cerebellum (**D**), midbrain (**E**), septal area (**F**), and cortex (**G**). Two-tailed Mann–Whitney test. The data are represented in a Tukey box plot (n = 9 per genotype except 8 in the cortex of WT). The data are given in **Figure 8—source data 1**. *p≤0.05, **p≤0.01, ***p≤0.001, and ns: not significant.

The online version of this article includes the following figure supplement(s) for figure 8:

**Source data 1.** Contrast-enhanced magnetic resonance imaging reveals a low level of tracer dispersion.

the slope was lower in the cerebellum, midbrain, and cortex (**Figure 8D, E and G**, **Figure 8—source data 1**). We therefore conclude that the absence of MLC1 impairs the intraparenchymal circulation and clearance of CSF.

## Developmental perivascular expression of MLC1 in the human cortex

In an attempt to move closer to the context of MLC in humans, we used immunohistochemical techniques to analyze the development of perivascular expression of MLC1 in human cortical sections from 15 weeks of gestation to 17 years of age (**Figure 9**, **Figure 9—source data 1**). Interestingly, perivascular MLC1 was detected as early as 15 weeks of gestation (**Figure 9A**, **Figure 9—source data 1**) and remained stable thereafter (**Figure 9B–F**, **Figure 9—source data 1**). These results suggested that in humans the MLC1/GlialCAM complex and the astrocyte's perivascular coverage are initiated prenatally.

## Discussion

*MLC1* (the main gene involved in MLC) encodes an astrocyte-specific protein located in perivascular astrocytic processes, where it forms a junctional complex with GlialCAM. Our previous research showed that, in the mouse, the MLC1/GlialCAM complex forms progressively after birth (between P5 and P15) (**Gilbert et al., 2019**). Our present work demonstrated for the first time that the P5–P15 time window is also important for the formation of perivascular astrocyte coverage. On P5, perivascular astrocytic processes covered only 50% of the vascular surface, and this coverage increased rapidly until completion on P15. We show that the absence of MLC1 impairs this developmental process together with the astrocytic perivascular processes' molecular maturation, with a lower Kir4.1 expression and a transient decrease in Aqp4 and Cx43 protein levels on P5. GlialCAM (whose anchorage in astrocytic membranes depends on MLC1) is distributed diffusely, as described previously (**Hoegg-Beiler et al., 2014**). From P10 onward, perivascular astrocytic processes incompletely cover the vessels and direct contact between neuronal components and the vessel wall are observed. Interestingly, the postnatal period is also an intense synaptogenic phase in the mouse brain (**Chung et al., 2015**); astrocytes and neurons might compete for the perivascular space during this time. In the absence of MLC1, neurons might either stabilize or insert into the space left free by perivascular astrocytic processes.

Remodeling of the gliovascular interface is accompanied by a loss of astrocyte polarity in *Mlc1* KO. Firstly, the perivascular distribution of Cx43 gap junction at the perivascular surface is modified with more and larger puncta, and the perivascular astrocytic processes are delineated by large gap junction plaques. Interestingly, disorganization of Cx43 was recently reported in *GlialCAM* KO mice (**Baldwin et al., 2021**); this supports the hypothesis whereby the MLC1/GlialCAM complex organizes gap junction coupling in perivascular astrocytic processes. Secondly, *Mlc1* KO astrocytes have an abnormally high number of ramifications, which tend to project toward the blood vessels. Thirdly, there are changes in the perivascular astrocytic processes' polarity and organization (stacking, interpenetration, and the presence of basal lamina on the parenchymal side). Lastly, astrocytic perivascular processes are less mechanically cohesive. It was recently demonstrated that the astrocyte arborization becomes more complex between P7 and P21 – the period during which the perivascular MLC1/GlialCAM astrocytic complex forms (**Clavreul et al., 2019**; **Gilbert et al., 2019**). The formation of perivascular astrocytic process coverage (mediated by the MLC1/GliaCAM complex) might influence this process and thus polarize astrocytes. Taken as a whole, our data demonstrate that MLC1 is crucial for the development of astrocyte morphology, perivascular polarity, and perivascular coverage.

What impact, then, does the lack of MLC1 have on the gliovascular physiology? Although astrocytes are key regulators of the blood–brain barrier's integrity (**Abbott et al., 2006**; **Alvarez et al., 2013**; **Castro Dias et al., 2019**; **Cohen-Salmon et al., 2021**), the incomplete perivascular astrocyte

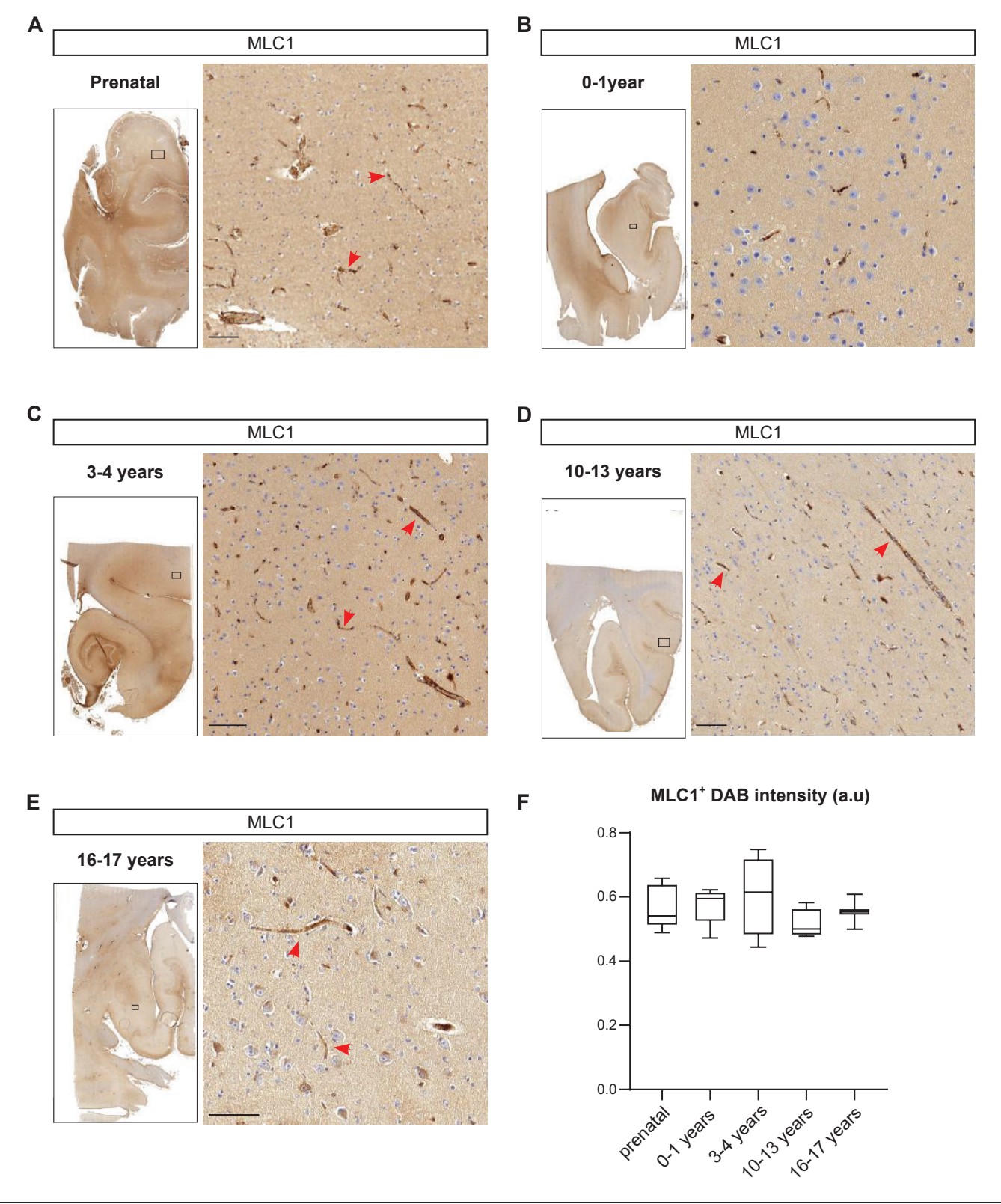

**Figure 9.** Developmental perivascular expression of MLC1 in the human cortex. (**A–E**) Representative images of MLC1-immunostained human cortical slices (left) and a higher magnification image of the parenchyma in the boxed areas (right) at the prenatal stage (weeks of gestation: 15; 21; 28; 30; 39) (**A**); 0–1 year of age (ages: 3 weeks; 1 month; 2 months; 3 months; 8 months; 1 year) (**B**); 3–4 years of age (3 years; 4 years (n = 2)) (**C**); 10–13 years of age (10 years; 11 years; 12 years; 13 years (n = 2)) (**D**); and 16–17 years of age (16 years; 17 years) (**E**). Scale bar: 100 μm. MLC1 immunostaining (arrowheads)

*Figure 9 continued on next page*

*Figure 9 continued*

was revealed with 3, 3'-diaminobenzidine (DAB). (**F**) (DAB) intensity was quantified and presented as a Tukey box plot. We applied the Kruskal–Wallis test (overall, in bold) and a two-tailed Mann–Whitney test (for comparing stages). The number of samples per developmental age was 5 for the prenatal samples, 5 for 0–1 years of age, 4 for 3–4 years of age, 4 for 10–13 years of age, and 2 for 16–17 years of age. The data are given in *Figure 9—source data 1*.

The online version of this article includes the following figure supplement(s) for figure 9:

**Source data 1.** Developmental perivascular expression of MLC1 in the human cortex.

coverage, the abnormal astrocytic polarity, and the loss of the perivascular astrocytic processes' mechanical cohesiveness did not alter blood–brain barrier's integrity in *Mlc1* KO. Nevertheless, these changes probably alter the 'barrier' formed by perivascular astrocytic processes around the vessels. In turn, this might greatly affect perivascular homeostasis and astrocyte signaling toward the vascular compartment (*Abbott et al., 2006*; *Yao et al., 2020*). The astrocytes' perivascular organization is thought to be crucial for regulating the fluxes of CSF and interstitial fluid into the parenchyma (*Abbott et al., 2018*). Perivascular astrocytic process alterations in *Mlc1* KO mice might therefore be directly linked to the loss of DOTA-Gd intraparenchymal drainage observed. By reducing the volume of subarachnoid spaces, megalencephaly in *Mlc1* KO mice may also alter the brain's drainage capacity, as suggested by the lower level of DOTA-Gd transport into the CSF observed in *Mlc1* KO mice. In the presence of an intact blood–brain barrier, impaired parenchymal circulation of the CSF/interstitial fluid might (i) contribute to the fluid accumulation and megalencephaly (*Dubey et al., 2015*; *van der Knaap et al., 2012*), (ii) lead to the progressive accumulation of harmful molecules in the brain, and (iii) thus increase susceptibility to neural disorders (*Rasmussen et al., 2018*).

Epilepsy is a significant component of MLC (*Yalçinkaya et al., 2003*). *Mlc1* KO mice display spontaneous epileptiform activity, an abnormally low threshold for seizure induction, and an elevated extracellular potassium concentration in the hippocampus upon prolonged high-frequency stimulation (*Dubey et al., 2018*). The absence of Kir4.1 leads to hyperexcitability and epilepsy (*Sibille et al., 2014*). The low level of perivascular Kir4.1 observed in *Mlc1* KO mice might link MLC to epilepsy.

Unlike skeletal or cardiac muscle, SMCs are not terminally differentiated and are extremely plastic. They constantly integrate signals from their local environment and undergo profound phenotypic changes in response to variations in their local environment (*Owens, 1995*; *Owens et al., 2004*). By perturbing perivascular homeostasis, the abnormal development of perivascular astrocytic processes in the absence of MLC1 might affect the postnatal acquisition of vascular smooth muscle cells' contractile properties and thus result in hypoperfusion and defective neurovascular coupling. This hypothesis is supported by the fact that perivascular astrocyte coverage, MLC1 expression (*Gilbert et al., 2019*), and vascular smooth muscle cell contractile differentiation (*Slaoui et al., 2021*) develop concomitantly. Interestingly, deletion of astrocytic laminin γ1 was shown to lead to the loss of vascular smooth muscle cell contractile proteins (*Chen et al., 2013*), which indicated a functional link between astrocytes and vascular smooth muscle cells. Our data now suggest that astrocytes have a critical role in the postnatal differentiation of contractile vascular smooth muscle cells.

Alteration of vascular smooth muscle cell contractility might influence brain perfusion and neurovascular coupling, which are critical for oxygen and nutrient delivery to neurons (*Iadecola, 2017*) this impairment might compromise neuronal and cerebral functions. Furthermore, the maintenance of axonal myelination makes extraordinary metabolic demands on oligodendrocytes (*Harris and Attwell, 2012*; *Rosko et al., 2019*). The lack of vascular smooth muscle cell contractility and the changes in blood perfusion and cerebral blood flow regulation resulting from the absence of MLC1 might contribute to progressive intramyelinic edema. This hypothesis is also supported by the fact that myelin vacuolation in *Mlc1* KO mice does not start until the age of 3 months (*Dubey et al., 2015*; *Hoegg-Beiler et al., 2014*). Finally, alteration of vascular smooth muscle cell contractility could affect circulation of the CSF and perivascular fluid, although the heart beat is the main driver (*Iliff et al., 2013*).

Together, the development of perivascular astrocytic process, vascular smooth muscle cell maturation, CSF flux, and cerebral blood flow defects precede myelin vacuolation in *Mlc1* KO mice and so are probably primary events in the pathogenesis of MLC. Our observation of swollen perivascular astrocytic processes on 1 year but not on P60 *Mlc1* KO mice indicates that edema develops progressively in the absence of MLC1 and suggests that the pathogenic process is irresistible.

In contrast to the mouse, expression of MLC1 and the astrocyte's perivascular coverage are initiated prenatally in humans, as already suggested by earlier observations of perivascular GFAP and AQP4 expression (*El-Khoury et al., 2006*). These results suggest that in humans impaired perivascular astrocytic process formation linked to the absence of MLC1 might occur prenatally, deregulating perivascular homeostasis and then the perinatal differentiation of vascular smooth muscle cell contractility (*Gilbert et al., 2019*; *Slaoui et al., 2021*), leading to a progressive white matter vacuolation.

In conclusion, we showed that the astrocyte-specific protein MLC1, the absence of which causes MLC, is critical for the postnatal development of perivascular astrocyte coverage, the acquisition of vascular smooth muscle cell contractility, and parenchymal CFS/ISF (cerebrospinal fluid/interstitial fluid) efflux. Our results shed light on the role of astrocytes in the postnatal acquisition of vascular smooth muscle cell contractility, a crucial component of neurovascular coupling in the brain. Our data indicate that MLC could be considered primarily as an early developmental disorder of the gliovascular unit. Moreover, our study illustrates how looking at physiopathological processes in a rare disease can enlighten important aspects of the brain's physiology.

# Materials and methods

## Key resources table

| Reagent type (species) or resource | Designation | Source or reference | Identifiers | Additional information |
|---|---|---|---|---|
| Sequence-based reagent | Acta2_F | This study | PCR primers | GTCCCAGACATCAGGGAGTAA |
| Sequence-based reagent | Acta2_R | This study | PCR primers | TCGGATACTTCAGCGTCAGGA |
| Sequence-based reagent | Atp1b1_F | This study | PCR primers | GCTGCTAACCATCAGTGAACT |
| Sequence-based reagent | Atp1b1_R | This study | PCR primers | GGGGTCATTAGGACGGAAGGA |
| Sequence-based reagent | Mdr1a (Abcb1)_F | This study | PCR primers | GATAGGCTGGTTTGATGTGC |
| Sequence-based reagent | Mdr1a (Abcb1)_R | This study | PCR primers | TCACAAGGGTTAGCTTCCAG |
| Sequence-based reagent | Cldn5_F | This study | PCR primers | TAAGGCACGGGTAGCACTCA |
| Sequence-based reagent | Cldn5_R | This study | PCR primers | GGACAACGATGTTGGCGAAC |
| Sequence-based reagent | Gapdh_F | This study | PCR primers | AGGTCGGTGTGAACGGATTTG |
| Sequence-based reagent | Gapdh_R | This study | PCR primers | TGTAGACCATGTAGTTGAGGTCA |
| Antibody | Claudin 5 (rabbit polyclonal) | Thermo Fisher | 34-1600 | Western blot (1:500) |
| Antibody | H3 (mouse monoclonal) | Ozyme | 14269S | Western blot (1:2000) |
| Antibody | P-gP (mouse monoclonal) | Enzo | ALX-801-002C100 | Western blot (1:200) |
| Antibody | SMA_cy3 (mouse monoclonal) | Sigma | C6198 | Immunofluorescence (1:250); western blot (1:1000) |
| Antibody | Pecam-1 (goat polyclonal) | R&D Systems | AF3628 | Immunofluorescence (1:300) |
| Antibody | Connexin 43 (mouse monoclonal) | BD transduction | 610062 | Immunofluorescence (1:200); western blot (1:500) |
| Antibody | GlialCAM (rabbit polyclonal) | Provided by Raul Estevez, Universitat de Barcelona, Spain. | / | Immunofluorescence (1:500); western blot (1:500) |

*Continued on next page*

*Continued*

| Reagent type (species) or resource | Designation | Source or reference | Identifiers | Additional information |
|---|---|---|---|---|
| Antibody | Aquaporin 4 (rabbit polyclonal) | Sigma | A5971 | Immunofluorescence (1:500); western blot (1:500) |
| Antibody | Neurofilament motor N (mouse monoclonal) | Provided by Beat M. Riederer, University of Lausanne, Switzerland. | / | Immunofluorescence (1:40); western blot (1:40) |
| Antibody | GFAP (rabbit polyclonal) | Sigma | G9269 | Immunofluorescence (1:500) |
| Antibody | MLC1human (rabbit polyclonal) | Provided by Raul Estevez, Universitat de Barcelona, Spain. | / | Immunofluorescence (1:200) |
| Antibody | MLC1pan (rabbit polyclonal) | Provided by Raul Estevez, Universitat de Barcelona, Spain. | / | Western blot (1:500) |
| Antibody | Kir 4.1 (rabbit polyclonal) | Alomone Labs | APC-035 | Western blot (1:5000) |
| Antibody | Kir 4.1 extracellular (rabbit polyclonal) | Alomone Labs | APC-165 | Immunofluorescence (1:200) |
| Antibody | Anti-rabbit_alexa 555 (goat polyclonal) | Thermo Fisher | A21429 | Immunofluorescence (1:2000) |
| Antibody | Anti-rabbit_alexa 488 (goat polyclonal) | Thermo Fisher | A11034 | Immunofluorescence (1:2000) |
| Antibody | Anti-mouse_alexa 555 (goat polyclonal) | Thermo Fisher | A21424 | Immunofluorescence (1:2000) |
| Antibody | Anti-oat_alexa 647 (donkey polyclonal) | Thermo Fisher | A-21447 | Immunofluorescence (1:2000) |
| Antibody | Anti-rabbit_HRP (goat polyclonal) | Biovalley | CSA2115 | Western blot (1:2500) |
| Antibody | Anti-mouse_HRP (goat polyclonal) | Biovalley | CSA2108 | Western blot (1:2500) |
| Antibody | Anti-rabbit STAR RED (goat polyclonal) | Abberior | STRED-1002 | Immunofluorescence (1:200) |
| Antibody | Anti-mouse STAR RED (goat polyclonal) | Abberior | STRED-1001 | Immunofluorescence (1:200) |
| Other | Alexa-conjugated isolectin (*Griffonia simplicifolia*) 647 | Thermo Fisher | I32450 | Immunofluorescence (1:100) |
| Other | Alexa-conjugated isolectin (*Griffonia simplicifolia*) 594 | Thermo Fisher | I21413 | Immunofluorescence (1:100) |

## Animals

All animal experiments were carried out in compliance with the European Directive 2010/63/EU on the protection of animals used for scientific purposes and the guidelines issued by the French National Animal Care and Use Committee (reference: 2019021814073504 and 2019022113258393). *Mlc1* KO mice were maintained on a C57BL6 genetic background (*Hoegg-Beiler et al., 2014*).

## Brain microvessel purification

Microvessels were isolated from whole brain using selective filtration, as described previously (*Boulay et al., 2015*). We purified vessels that passed through 100 µm pores but not 20 µm pores (*Boulay et al., 2015*). Brain vessels from two animals were pooled for the 2-month samples, three for P15 samples and five for P5 samples.

## Immunohistochemical analysis

### Brain slices

Mice were anesthetized with pentobarbital (600 mg/kg, i.p.) and killed by transcardiac perfusion with phosphate buffered saline (PBS)/paraformaldehyde (PFA) 4%. The brain was removed and cut into 40-µm-thick sections using a Leitz (1400) cryomicrotome.

### Purified microvessels

Microvessels were plated on a glass slide coated with Cell Tak (Corning, NY) and fixed in PBS/PFA 4% for 15 min at room temperature (RT).

Brain slices or microvessels on glass slides were immersed in the blocking solution (PBS/normal goat serum (NGS) 5%/Triton X-100 0.5%) for 1 hr at RT and then incubated with primary antibodies (see the Key resources table) diluted in the blocking solution 12 hr at 4°C. After three washes in PBS, slices were incubated for 2 hr at RT (or overnight for STED experiments) with secondary antibodies and Hoechst dye, rinsed in PBS, and mounted in Fluormount G (Southern Biotech, Birmingham, AL) for confocal analysis or Abberior Mount solid antifade medium for STED imaging.

Tissues were imaged using a 40× objective on a Zeiss Axio-observer Z1 with a motorized XYZ stage (Zeiss, Oberkochen, Germany). For STED imaging, we used a STEDyCON microscope (Abberior Instruments, Göttingen, Germany) with a 100×/1.46 Plan-ApoChromat DIC Oil (Zeiss). Alexa 594 and Star-red fluorescence was depleted with a laser at 775 nm. The pixel size was set to 25 nm, with a 1.13 pinhole.

## Quantification of immunofluorescence

Images were analyzed using ImageJ/Fiji software (*Schindelin et al., 2012*; *Schneider et al., 2012*).

### Microvessels

The isolectin B4 channel was first processed with a *subtract background* filter (rolling ball radius = 50 pixels), a *Gaussian Blur* filter (sigma = 10 pixels), and using the *Tubness* ImageJ plugin (sigma = 1). The resulting image was converted to a mask using the *Huang dark* threshold, which was dilated three times. A distance map image of this mask was created using *local Thickness* (threshold = 1), and the surface area of isolectin-B4-positive vessels was measured. Aqp4 or NF-M channels were processed by *Bleach correction* using the *simple ratio* method. A *subtract background* filter (rolling ball radius = 50 pixels) and a *Median* filter (radius = 2 pixels) were then applied. The resulting picture was converted to a mask using the *Default dark* threshold. A third mask for the Aqp4 or NF-M signal contained in the isolectin B4 channel was created by using the *imageCalculator AND* method to combine the two previously created masks. We calculated the surface ratio of this third mask/isolectin B4.

### STED

Vessels with a diameter below 15 µm were analyzed. For Aqp4, six individual $2 \times 0.1$ µm$^2$ surfaces perpendicular to the vessel wall were drawn half inside the vessel and half outside, starting from the contact zone between the Aqp4 staining and the isolectin B4 staining. We then calculated the Aqp4/IB4 ratio. For Cx43, GlialCAM, and Kir4.1, the particles' size and number were quantified in a 1 µm perivascular area (the vessel's boundary was based on the isolectin B4 staining). We applied a *Gaussian filter* (sigma = 1), with an *Intermodes dark* threshold for Cx43 and GlialCAM and a *Moments dark* threshold for Kir 4.1.

## Western blots

Proteins were extracted from one brain hemisphere or from purified microvessels in 2% SDS (500 µL or 50 µL per sample, respectively) with EDTA-free Complete Protease Inhibitor (Roche), sonicated three times at 20 Hz (Vibra cell VCX130) and centrifuged for 20 min at 10,000 *g* at 4°C. Supernatants were heated in Laemmli loading buffer for 5 min at 56°C. Proteins were extracted from one brain hemisphere per sample in 500 µL SDS 2% under the same conditions. The protein content was measured using the Pierce 660 nm protein assay (Thermo Scientific, Waltham, MA). Equal amounts of proteins were separated by denaturing electrophoresis on Mini-Protean TGX stain-free gels (Bio-Rad) and then electrotransferred to nitrocellulose membranes using the Trans-blot Turbo Transfer System

(Bio-Rad). Membranes were hybridized, as described previously (*Ezan et al., 2012*). The antibodies used in this study are listed in the Key resources table. Horseradish peroxidase activity was visualized using enhanced chemiluminescence in a Western Lightning Plus system (Perkin Elmer, Waltham, MA). Chemiluminescent imaging was performed on a FUSION FX system (Vilber, South Korea). At least four independent samples were analyzed in each experiment. The level of chemiluminescence for each antibody was normalized against that of histone H3 or a stain-free membrane (enabling bands to be normalized against the total protein on a blot).

## Quantitative RT-PCR

RNA was extracted using the Rneasy Lipid Tissue Mini Kit (Qiagen, Hilden, Germany). cDNA was then generated using the Superscript III Reverse Transcriptase Kit (Thermo Fisher). Differential levels of cDNA expression were measured using droplet digital PCR. Briefly, cDNA and primers (see the Key resources table) were distributed into approximately 10,000–20,000 droplets. cDNAs were then PCR-amplified in a thermal cycler and read (as the number of positive and negative droplets) with a QX200 Droplet Digital PCR System (Bio-Rad). The ratio for each tested gene was normalized against the total number of positive droplets for *Gapdh*.

## In situ brain perfusion

Mice were anesthetized with ketamine-xylazine (140 and 8 mg/kg, respectively, i.p.), and a polyethylene catheter was inserted into the carotid veins. The heart was incised, and the perfusion was started immediately (flow rate: 2.5 mL/min) so as to completely replace the blood with Krebs carbonate-buffered physiological saline (128 mM NaCl, 24 mM $NaHCO_3$, 4.2 mM KCl, 2.4 mM $NaH_2PO_4$, 1.5 mM $CaCl_2$, 0.9 mM $MgCl_2$), (9 mM D-glucose) supplemented with [$^{14}$C] sucrose (0.3 µCi/mL) (Perkin Elmer Life Sciences, Courtaboeuf, France) as a marker of vascular integrity. The saline was bubbled with 95% $O_2$/5% $CO_2$ for pH control (7.4) and warmed to 37°C. Perfusion was terminated after 120 s by decapitating the mouse. The whole brain was removed from the skull and dissected out on a freezer pack. Brain hemisphere and two aliquots of perfusion fluid were placed in tared vials and weighed, digested with Solvable (Perkin Elmer) and mixed with Ultima gold XR (Perkin Elmer) for $^{14}$C dpm counting (Tri-Carb, Perkin Elmer). In some experiments, human serum albumin (40 g/L) (Vialebex, Paris, France) was added to the perfusion fluid in order to increase the hydrostatic pressure (~180 mmHg) and create shear stress (*Ezan et al., 2012*). The brain [$^{14}$C]-sucrose vascular volume (Vv, in µL/g) was calculated from the distribution of the [$^{14}$C]-sucrose: $Vv = X_v/C_v$, where $X_v$ (dpm/g) is the [$^{14}$C] sucrose concentration in the hemispheres and $C_v$ (dpm/µL) is the [$^{14}$C] sucrose concentration in the perfusion fluid (*Dagenais et al., 2000*). It should be noted that in mammals the very hydrophilic, low-molecular-weight (342 Da) disaccharide sucrose does not bind to plasma proteins and does not have a dedicated transporter. Accordingly, sucrose does not diffuse passively and thus serves as a marker of blood–brain barrier integrity (*Takasato et al., 1984*). In this context, variations in sucrose's distribution volume in the brain solely reflect changes in the blood–brain barrier's physical integrity.

## Vascular smooth muscle cell responsiveness

Mice were rapidly decapitated, and the brains were quickly removed and placed in cold (~4°C) artificial cerebrospinal fluid (aCSF) solution containing 119 mM NaCl, 2.5 mM KCl, 2.5 mM $CaCl_2$, 26.2 mM $NaHCO_3$, 1 mM $NaH_2PO_4$, 1.3 mM $MgSO_4$, 11 mM D-glucose (pH = 7.35). Brains were constantly oxygenated with 95% $O_2$–5% $CO_2$. Brain cortex slices (400 µm thick) were cut with a vibratome (VT2000S, Leica) and transferred to a constantly oxygenated (95% $O_2$–5% $CO_2$) holding chamber containing aCSF. Subsequently, individual slices were placed in a submerged recording chamber maintained at RT under an upright microscope (Zeiss) equipped with a CCD camera (Qimaging) and perfused at 2 mL/min with oxygenated aCSF. Only one vessel per slice was selected for measurements of vascular responsiveness at the junction between layers I and II of the somatosensory cortex and with a well-defined luminal diameter (10–15 µm). An image was acquired every 30 s. Each recording started with the establishment of a control baseline for 5 min. Vessels with an unstable baseline (i.e., a change in diameter of more than 5%) were discarded from analysis. Vasoconstriction was induced by the application of the thromboxane $A_2$ receptor agonist U46619 (9,11-dideoxy-11a,9a- epoxymethanoprostaglandin F2α, 50 nM, Sigma) for 2 min. The signal was recorded until it had returned to the baseline.

## Functional ultrasound

Two-month-old mice were anesthetized, and cerebral blood flow responses to whisker stimulation were determined using functional ultrasound imaging. The protocol is described in detail in *Anfray et al., 2019*. Briefly, mice were intubated and mechanically ventilated (frequency: 120 /min; tidal volume:10 mL/kg) by maintaining anesthesia with 2% isoflurane in 70% $N_2O$/30% $O_2$. Mice were placed in a stereotaxic frame, and the head was shaved and cleaned with povidone-iodine. An incision was made along the midline head skin (to expose the skull), and lidocaine spray was applied to the head. Whiskers on the left side were cut to a length of 1 cm. Anesthesia was switched to a subcutaneous infusion of medetomidine (Domitor, Pfizer, 0.1 mg/kg) and isoflurane, $N_2O$ and $O_2$ were withdrawn 10 min later. So that the isoflurane could dissipate and the cerebral blood flow could stabilization, the functional ultrasound measurements were initiated 20 min later. Ultrasound gel was applied between the ultrasound probe and the mouse's skull to ensure good acoustic coupling. The probe was positioned in the coronal plane, corresponding to the somatosensory barrel field cortex (S1bf; bregma –1.5 mm). Ultrafast acquisition was performed with an ultrasound sequence based on compounded plane wave transmission (11 angles from 10° to 10°, in increments of 2°) using a 15 MHz probe (Vermon, France; 100 µm × 100 µm in-plane pixels; slice thickness: 300 µm; elevation focus: 8 µm; frame rate: 500 Hz). The whiskers were mechanically stimulated three times for 30 s, interspaced with a 60 s rest period (total duration of the experiment: 300 s). Using MATLAB (the MathWorks Inc, Natick, MA), we calculated the coefficient for the correlation between the normalized power Doppler (PD) intensity over time and a step function following the stimulation pattern. An activation map was reconstructed by selecting only pixels with a correlation coefficient above 0.2. The relative PD increase was quantified as the mean PD signal in the activated area.

## Ultrasound localization microscopy

The acquisition and postprocessing steps for ultrasound localization microscopy were adapted from *Hingot et al., 2020*. For each image, 100 µL of Sonovue microbubbles were injected into the tail vein. Blocks of 800 compounded frames (–5° 0° 5°) at 1 kHz were acquired for 800 ms and saved for 200 ms; this scheme was repeated for 180 s. A combination of a Butterworth high-pass filter (second order, 20 Hz) and a singular value decomposition filter (10 values) was used to separate microbubble echoes from tissue echoes. The microbubbles' centroid positions were localized using a weighted average algorithm. Microbubbles were tracked through consecutive frames using MathWorks (the MathWorks Inc). The tracks were interpolated and smoothed using a five-point sliding window, and redundant positions were removed. A density image was reconstructed on an 11 µm × 10 µm grid.

## Magnetic resonance imaging

Magnetic resonance imaging was performed on a 7 T Pharmascan magnetic resonance imaging system (Bruker, Rheinstetten, Germany) equipped with volume transmit and surface receive coils and operated via Paravision 6.0 software (Bruker). An anatomical T2-weighted acquisition was performed prior to contrast injection, with the following parameters: echo time (TE) = 40 ms; repetition time (TR) = 3500 ms; flip angle (FA) = 90°; averages = 2; number of echoes = 8; on a 256 × 256 sagittal matrix with 20 contiguous 0.5-mm-thick slices (in-plan resolution = 0.07 × 0.07 mm) for a total duration of 2 min 41 s. The apparent diffusion coefficient was calculated from a multi-b diffusion-weighted echo planar imaging sequence, with the following parameters: TE = 35 ms; TR = 2000 ms; FA = 90°; number of segments = 4; 16 b-values = 20 $s.mm^{-2}$, 30 $s.mm^{-2}$, 40 $s.mm^{-2}$, 50 $s.mm^{-2}$, 75 $s.mm^{-2}$, 100 $s.mm^{-2}$, 150 $s.mm^{-2}$, 200 $s.mm^{-2}$, 300 $s.mm^{-2}$, 400 $s.mm^{-2}$, 500 $s.mm^{-2}$, 750 $s.mm^{-2}$, 1000$s.mm^{-2}$, 1250$s.mm^{-2}$, 1500$s.mm^{-2}$, 2000$s.mm^{-2}$; 12 directions; on a 128 × 40 sagittal matrix with nine contiguous 1-mm-thick slices (in-plan resolution: 0.15 × 0.15 mm), with use of a saturation band to remove the out-of-matrix signal, over a total duration of 25 min 45 s. The apparent diffusion coefficient acquisition was performed prior to contrast injection. To estimate the concentration of contrast agent, T1 maps before and after contrast injection were computed from a FAIR RARE (flow alternating inversion recovery, rapid acquisition with refocused echoes) acquisition derived from the Look-Locker T1 mapping sequence (*Karlsson and Nordell, 1999*), with the following parameters: TE = 5.3 ms; TR = 5000 ms; FA = 90°; RARE factor = 4; 10 inversion times (TI) = 10 ms, 21 ms, 44 ms, 195 ms, 410 ms, 862 ms, 1811 ms, 3807 ms, and 8000 ms; on a 64 × 64 single mediosagittal slice (in-plan resolution: 0.3 × 0.3 mm; thickness: 0.8 mm) for a total duration of 4 min 25 s. A single acquisition was performed before contrast

injection (to map the reference T1), and eight consecutive acquisitions were performed 10 min after contrast injection – providing dynamic data over 40 min.

## Injection of contrast agent into the CSF

1 µL of 500 mM DOTA-Gd (Dotarem, Guerbet, France) was injected over 1 min into the CSF with a glass micropipette through the cisterna magna, as described previously (*Gaberel et al., 2014*). Briefly, the mice were anesthetized with isoflurane (induction: 5%; maintenance: 2–3%) in 70% $N_2O$/30% $O_2$. The neck was shaved, and lidocaine was sprayed on for local analgesia. A vertical incision was performed, and the muscle planes were separated vertically upon reaching the cisterna magna. A micropipette formed from an elongated capillary glass tube and filled with 1 µL of DOTA-Gd was inserted into the cisterna magna. Before and after injection, 1 min pauses enabled the CSF pressure to normalize. Before micropipette removal, a drop of superglue was added to form a seal and prevent subsequent leakage of CSF. The incision was cleaned and then closed with 5.0 gauge surgical silk thread.

## Magnetic resonance imaging analyses

T1 values, quantitative contrast measurements, and apparent diffusion coefficient maps were calculated with in-house MATLAB code (R2021a, the MathWorks Inc; 2020). Regions of interest were determined using the Fiji image analysis suite (*Schindelin et al., 2012*).

T1 maps were computed from the FAIR RARE data. Briefly, T1 was extracted after fitting the signal recovery equation:

$$M_t = M_0 . exp\left(\frac{-t}{T1}\right)$$

The contrast agent concentration [CA] was determined from the equation

$$\left[CA\right] . r1 = \frac{1}{T1_{post}} - \frac{1}{T1_{pre}}$$

where r1, T1$_{post}$, and T1$_{pre}$ were respectively the T1 relaxivity, the T1 value after contrast, and the T1 value before contrast. The apparent diffusion coefficient was calculated as the slope of the log of signal loss for the b-value, according to the following equation:

$$ADC = \frac{ln\left(\frac{S_0}{S_b}\right)}{b}$$

where b = 1000s.mm$^{-2}$.

## Tissue clearing and immunohistochemical staining

Mice were killed with pentobarbital (600 mg/kg, i.p.). Brains were removed and post-fixed in 4% PFA for 24 hr at 4°C and then assessed using the 'immunolabeling-enabled three-dimensional imaging of solvent-cleared organs' technique (*Renier et al., 2014*). The samples were first dehydrated with increasingly concentrated aqueous methanol solutions (MetOH: 20, 40, 60, 80, and twice 100%, for 1 hr each) at RT and then incubated in 66% dichloromethane (DCM, Sigma-Aldrich)/33% MetOH overnight. After two washes in 100% MetOH, brains were incubated in 5% $H_2O_2$/MetOH overnight at RT, rehydrated with increasingly dilute aqueous methanol solutions (80, 60, 40, and 20%; 1 hr each). Before immunostaining, brains were permeabilized first for 2 × 1 hr at RT in 0.2% Triton X-100/PBS, for 24 hr at 37°C in 0.16% Triton X-100/2.3% glycine/20% DMSO/PBS, and then for 2 days at 37°C in 0.16% Triton X-100/6% donkey serum/10% DMSO/PBS. Brains were incubated for 3 days at 37°C with primary antibody diluted in a 0.2 Tween/1% heparin/3% donkey serum/5% DMSO/PBS solution, washed five times during 24 hr at 37°C in 0.2% Tween20/1% heparin/PBS solution, incubated for 3 days at 37°C with secondary antibody diluted in a 0.2 Tween/1% heparin/3% donkey serum/PBS solution, and another washed five times. The brain samples were then dehydrated again with a MetOH/$H_2O$ series (20, 40, 60, 80, and 100% for 1 hr each, and then 100% overnight) at RT. On the following day, brains were incubated for 3 hr in 66% DCM/33% MetOH and then twice for 15 min at RT in 100% DCM and lastly cleared overnight in dibenzyl ether.

The cleared tissues were imaged using a light sheet microscope and Inspector pro software (Lavision Biotec GmbH, Bielefeld, Germany). 3D reconstructions of the somatosensory cortex (a 400-µm-thick

column for Pecam-1 and 500–750 µm for SMA) were visualized with Imaris software (Bitplane). The length and number of branch points of Pecam-1- or SMA-immunolabeled brain vessels were quantified using the 'Surface' and 'Filament' tools in Imaris software (Oxford Instruments, Oxford). Anastomoses were measured by eye.

## Astrocyte morphology

Hippocampal slices were pictured using a 40× objective on a Zeiss Axio-observer Z1 with a motorized XYZ stage (Zeiss). To analyze astrocyte ramifications, we adapted a previously described technique (*Pannasch et al., 2014*). Using ImageJ software, seven concentric circles at 5 µm intervals were drawn around each astrocyte on confocal Z-stack images. The number of intersections of GFAP-positive astrocytic processes with each circle was counted.

We analyzed the astrocytes' orientation by adapting a previously described technique (*Ghézali et al., 2018*). Using ImageJ software, a grid delimiting 100 µm$^2$ squares was drawn on confocal Z-stack images oriented with the pyramidal cell layer or a vessel. The number of intersections of astrocytic GFAP-positive processes with horizontal lines (i.e., processes perpendicular to the pyramidal layer or vessel, so-called axial processes) and vertical lines (i.e., processes parallel to the pyramidal layer or vessel, so-called lateral processes) were counted. The cell's polarity index was defined as the ratio between the axial processes and the lateral processes. A polarity index of 1 indicates no polarity, whereas a polarity index greater than 1 indicates preferentially perpendicular orientation toward the pyramidal layer or the vessel.

## Electron microscopy

Mice were anesthetized with ketamine-xylazine (140 and 8 mg/kg, respectively, i.p.) and transcardially perfused with the fixative (2% PFA, 3% glutaraldehyde, 3 mM CaCl$_2$ in 0.1 M cacodylate buffer pH 7.4) for 12 min. The brains were removed and left overnight at 4°C in the same fixative. Brain fragments (0.3 × 1 × 1 mm$^3$) were postfixed first in 0.1 M cacodylate buffer pH 7.4 + 1% OsO$_4$ for 1 hr at 4°C and then in 1% aqueous uranyl acetate for 2 hr at RT. After dehydration in graded ethanol and then propylene oxide, the fragments were embedded in EPON resin (Electron Microscopy Sciences, Hatfield, PA). Ultrathin (80 nm) sections were prepared, stained with lead citrate, and imaged in a Jeol 100S transmission electron microscope (Jeol, Croissy-sur-Seine, France) equipped with a 2k × 2k Orius 830 CCD camera (Roper Scientific, Evry, France). Cells and structures were identified as follows. Endothelial cells are thin, elongated cells lining the vessel lumen and which can be joined together by electron-dense tight junctions. The endothelial cells are surrounded by a continuous layer of acellular matrix (the basal lamina). In the normal adult brain, the vascular wall is totally surrounded by astrocyte perivascular processes, which can be recognized by the presence of thin intermediate filaments and by the absence of basal lamina on the parenchymal side. Neuronal structures are typically recognized by the presence of microtubules, vesicles in presynapses, and an electron-dense region between the pre- and postsynapses.

## Human tissue immunohistochemistry

Our study included specimens obtained from the brain collection 'Hôpitaux Universitaires de l'Est Parisien – Neuropathologie du développement' (Biobank identification number BB-0033-00082). Informed consent was obtained for autopsy of the brain and histological examination. Our study included fetal brains obtained from spontaneous or medical abortion that did not display any significant brain pathology. After removal, brains were fixed with formalin for 5–12 weeks. Macroscopic analysis was performed to select samples that were embedded in paraffin, sliced in 7 µm sections and stained with hematein for a first histological analysis. Immunohistochemical analyses were performed on coronal slices that included the temporal telencephalic parenchyma and hippocampus. They were dewaxed and rinsed before incubation in citrate buffer (pH 9.0). Expression of MLC1 on the sections was detected using the Bond Polymer Refine Detection Kit (Leica) with specific antibodies and an immunostaining system (Bond III, Leica). Images were acquired using a slide scanner (Lamina, Perkin Elmer). Staining was analyzed using QuPath (*Bankhead et al., 2017*). A QuPath pixel classifier was trained to discriminate between DAB-positive spots and background areas. We selected a pixel classifier that used a random trees algorithm and four features: a Gaussian filter to select intensity, and the three structure tensor eigenvalues to select thin elongated objects. The classifier was trained on

manually annotated MLC1 spots and the background area on one image per developmental stage. When the result was satisfactory, the pixel classifier was used to detect MLC1 in selected regions of interest.

## Statistics

For all variables, the normality of data distribution was probed using the Shapiro–Wilk test before the appropriate statistical test was chosen. Test names and sample sizes are indicated in the figure legends. Detailed results are presented in the figure source data files.

## Acknowledgements

We are grateful to the donors who support the charities and charitable foundations cited below. This work was funded by grants from the *Association Européenne contre les Leucodystrophies* (ELA, grant reference ELA2012-014C2B), the *Fondation pour la Recherche Médicale* (FRM, grant reference AJE20171039094), and the *Fondation Maladies Rares* (grant reference 20170603). A Gilbert's PhD was funded by the FRM (grant reference: PLP20170939025p60) and ELA (grant reference: ELA2012-014C2B). The creation of the Center for Interdisciplinary Research in Biology (CIRB) was funded by the 'Fondation Bettencourt Schueller.' We thank Fawzi Boumezbeur, Aloïse Mabondzo, Corinne Blugeon, Laurent Jourdren, and Stéphane Le Crom for helpful discussions. We thank Isabelle Bardou for her help writing ethical documents. We thank Louise Charpentier and Ines Masurel for their help in analyzing astrocyte morphology and polarity. Lastly, we thank Virginie Mignon for help with transmission electron microscopy, Julien Dumont for help with STED imaging, and Augustin Walter for help with confocal image analysis. Copyediting assistance was provided by Biotech Communication SARL (Ploudalmézeau, France) and Life Science Editors. Despite our efforts, our work has not received any funding from the French National Agency for Research (ANR).

## Additional information

### Funding

| Funder | Grant reference number | Author |
|---|---|---|
| Association Européenne contre les Leucodystrophies | ELA2012-014C2B | Martine Cohen-Salmon |
| Fondation Maladies Rares | 20170603 | Martine Cohen-Salmon |
| Fondation pour la Recherche Médicale | PLP20170939025p60 | Alice Gilbert |
| Fondation pour la Recherche Médicale | AJE20171039094 | Martine Cohen-Salmon |

The funders had no role in study design, data collection and interpretation, or the decision to submit the work for publication.

### Author contributions

Alice Gilbert, Xabier Elorza-Vidal, Data curation, Formal analysis, Investigation, Methodology, Validation, Visualization; Armelle Rancillac, Audrey Chagnot, Mervé Yetim, Data curation, Investigation, Methodology, Validation, Visualization; Vincent Hingot, Investigation, Methodology, Supervision, Validation, Visualization; Thomas Deffieux, Isabelle Brunet, Mickael Tanter, Methodology, Supervision; Anne-Cécile Boulay, Investigation, Methodology; Rodrigo Alvear-Perez, Virginie Mignon, Investigation; Salvatore Cisternino, Conceptualization, Formal analysis, Investigation, Methodology, Validation, Visualization; Sabrina Martin, Sonia Taïb, Methodology; Aontoinette Gelot, Resources; Maryline Favier, Investigation, Methodology, Validation, Visualization; Xavier Declèves, Funding acquisition, Methodology, Supervision; Raul Estevez, Methodology, Resources; Denis Vivien, Conceptualization, Funding acquisition, Methodology, Supervision, Validation, Visualization; Bruno Saubaméa, Conceptualization, Investigation, Methodology, Supervision, Validation, Visualization; Martine Cohen-Salmon,

Conceptualization, Data curation, Formal analysis, Funding acquisition, Investigation, Methodology, Supervision, Validation, Visualization, Writing - original draft, Writing - review and editing

**Author ORCIDs**
Armelle Rancillac http://orcid.org/0000-0003-1085-5929
Anne-Cécile Boulay http://orcid.org/0000-0001-5620-6209
Salvatore Cisternino http://orcid.org/0000-0001-8500-3574
Sonia Taïb http://orcid.org/0000-0002-9981-5204
Isabelle Brunet http://orcid.org/0000-0002-5490-2937
Raul Estevez http://orcid.org/0000-0003-1579-650X
Martine Cohen-Salmon http://orcid.org/0000-0002-5312-8476

**Ethics**
Human subjects: Our study included specimens obtained from the brain collection "Hôpitaux Universitaires de l'Est Parisien - Neuropathologie du développement" (Biobank identification number BB-0033-00082). Informed consent was obtained for autopsy of the brain and histological examination. All animal experiments were carried out in compliance with the European Directive 2010/63/EU on the protection of animals used for scientific purposes and the guidelines issued by the French National Animal Care and Use Committee (reference: 2019021814073504 and 2019022113258393).

**Decision letter and Author response**
Decision letter https://doi.org/10.7554/eLife.71379.sa1
Author response https://doi.org/10.7554/eLife.71379.sa2

## Additional files

**Supplementary files**
• Transparent reporting form

**Data availability**
All data has been included in the manuscript and supporting files (please see supplementary files for raw data).

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
