## [Editor Report]

In this manuscript by Gilbert et al., the authors investigate how deletion of astrocytic membrane protein, MLC1, causes a rare disease called megalencephalic leukoencephalopathy with subcortical cysts (MLC). Through multiple experimental approaches, the authors show that Mlc1 knock-out mice exhibit defects in postnatal maturation of perivascular astrocyte coverage as well as vascular smooth muscle cell contractility, neurovascular coupling, and parenchymal CSF flow. Together, this work provides important frame works that MLC is caused by defects in the development of gliovascular unit.

---

## [Decision Letter]

**Decision letter after peer review:**

Thank you for submitting your article "MLC1 sustains the postnatal development of perivascular astrocytic processes: Insights into the pathogenesis of Megalencephalic leukoencephalopathy with subcortical cysts" for consideration by *eLife*. Your article has been reviewed by 3 peer reviewers, including Won-Suk Chung as Reviewing Editor and Reviewer #1, and the evaluation has been overseen by Didier Stainier as the Senior Editor. The following individuals involved in review of your submission have agreed to reveal their identity: Vincent Prevot (Reviewer #2); Injune Kim (Reviewer #3).

Essential revisions:

1) The authors reported that Mlc1 KO mice have more neuronal processes in contact with the vessels (Figure 6), and astrocytic processes appears to grow toward the vessels as well (Figure 5). However, they also suggested that the cohesiveness of PvAPs and the associated neuronal fibers to the vessels could be impaired based on the experiment from the mechanical purification of MVs (Figure 4). Since the interactions between PvAPs, neuronal fibers and the vessel are critical in understanding the neurovascular unit, the reduced cohesiveness of PvAPs and the associated neuronal fibers to the vessel in Mlc1 KO brains should be validated with additional experimental approach. Also, the authors need to discuss this point in more detail.

2) It is interesting that DOTA-Gd tracer shows different traces in Mlc1 KO brains. However, it is unclear how MLC1 deletion affects glymphatic system with different degrees in various brain regions. Does the tracer normally enter to the perivascular spaces in Mlc KO brains? Does the tracer leak out more from the perivascular spaces in Mlc1 KO mice? Is the general clearance or drainages of the tracer impaired in Mlc1 KO mice? Would these defects be originated by the reduced perivascular astrocyte coverage or the reduced vasoconstriction itself? Addressing these questions will strengthen the potential implication of Mlc1 in the circulation and clearance of CSF.

3) The defective polarity of astrocytes should be better described by using other markers other than GFAP. The distribution of Aquaporin4, Cx43 or several glutamate transporters in the specific compartment of astrocytes can be examined.

*Reviewer #1 (Recommendations for the authors):*

1. In general, Western blot should be presented with control loading.

2. Figure 5 and 8, as well as Figure 6 and 9 are almost identical. I suggest either combining these figures into two main figures or send the mirror figures to supplementary figures.

3. In Figure 4B, why Glialcam western showed non-significant changes at P15 and P60 although IHC data showed Glialcam is not expressed at all in MLlc1 KO astrocytes?

4. In Figure 4C, Aquaporin4 western blot seems to be just copied from the previous western blot data from Figure 4B.

*Reviewer #2 (Recommendations for the authors):*

This very interesting manuscript by Gilbert and colleagues uncovers that the astrocyte specific membrane protein MLC1, the mutation of which causes a rare disease called megalencephalic leukoencephalopathy with subcortical Cysts (MLC), plays a fundamental role in the postnatal development of the gliovascular unit and the organization of the perivascular astrocyte processes, in particular. To reach this conclusion, the authors used an elegant multiscale approach including in vivo MRI, in vivo functional ultrasound, ex vivo analysis of vascular constriction, anatomical approaches at the light and electron microscopic level, and molecular characterization of the gliovascular unit from isolated microvessels. The manuscript is very well-written although it uses too many (unnecessary) abbreviations, which prevents a fluid reading of the manuscript, results are well illustrated and convincing and the discussion is reasonable. The paper certainly is an important piece of work, which would feature well in *eLife*.

I have a major concern regarding the results reported in Figure 4D, which seem somewhat contradictory to those shown in Figure 6A-F. Indeed, the authors report in Figure 4D that there is less Neurofilament-M protein around isolated microvessels in MLC1 KO mice, whereas Figure 6A-F shows that these animals have more neuronal processes in contact with the vessels than in wiltypes. How can the authors explain this?

*Reviewer #3 (Recommendations for the authors):*

Astrocytes unique to the central nervous system such as the brain is known to play a role in maintaining BBB integrity by surrounding brain vessels tightly vis their end-feet structure. However, how astrocytes form end-feet along brain vessels is largely unknown. The present manuscript by Gilbert et al., describes how perivascular astrocytic processes are established during postnatal development and pinpoint a player involved. They find that MLC1 deletion resulted in disorganized perivascular astrocytic processes and defective contractility of vascular smooth muscle cells, thereby leading to impaired neurovascular coupling and intraparenchymal fluid clearance. To assess the functions on top of anatomy, they monitored blood perfusion and fluid transport of CSF in vivo using ultrasound imaging and MRI. I think that only a few issues remain to be clarified publication of this manuscript.

1) In Figure 7D and 7G, but not in 7E and 7F, the basal values are significantly high in Mlc1 KO mice at 10 min after Gd infusion. How is the pattern of basal Gd concentration different depending on the brain area?

2) Although endothelial cells look normal in immunofluorescence images and TEM images, it needs to be clarified whether the function of BBB is affected by assessing leakage of tracers infused into blood.

---

## [Author Response]

Essential revisions:1) The authors reported that Mlc1 KO mice have more neuronal processes in contact with the vessels (Figure 6), and astrocytic processes appears to grow toward the vessels as well (Figure 5). However, they also suggested that the cohesiveness of PvAPs and the associated neuronal fibers to the vessels could be impaired based on the experiment from the mechanical purification of MVs (Figure 4). Since the interactions between PvAPs, neuronal fibers and the vessel are critical in understanding the neurovascular unit, the reduced cohesiveness of PvAPs and the associated neuronal fibers to the vessel in Mlc1 KO brains should be validated with additional experimental approach. Also, the authors need to discuss this point in more detail.

To strengthen our analysis, we now give the results of a parallel quantitative immunofluorescence analysis of purified brain vessels (presented in a new figure, Figure 5). The results show that the part of the Aqp4 and NF-M perivascular immunolabeling is absent in the Mlc1 KO. Taken as a whole, our data demonstrate that PvAPs and the associated neuronal fibers (which normally remain attached to brain vessels during mechanical purification) are lost during the purification process in Mlc1 KO mice but not in the WT. In conclusion, the absence of MLC1 reduces the mechanical cohesiveness of PvAPs and the associated neuronal fibers.

2) It is interesting that DOTA-Gd tracer shows different traces in Mlc1 KO brains. However, it is unclear how MLC1 deletion affects glymphatic system with different degrees in various brain regions. Does the tracer normally enter to the perivascular spaces in Mlc KO brains? Does the tracer leak out more from the perivascular spaces in Mlc1 KO mice? Is the general clearance or drainages of the tracer impaired in Mlc1 KO mice?

Paravascular transport (as revealed by the injection of a tracer into the CSF) depends mainly on dispersion of the tracer in the subarachnoid space (SAS), the cisternae, and the parenchyma (including the interstitial and perivascular spaces). The uneven, slow dispersion of the tracer within the SAS means that the tracer’s kinetics in the parenchyma are region-dependent. These differences can be accentuated by regional differences in the anatomy of the brain’s vasculature, i.e. the presence or absence of a perivascular space and the vessel’s topology. Lastly, the amount of DOTA-Gd available for diffusion within the parenchyma depends directly on its local concentration in the SAS. This can be seen on our contrast concentration maps (see Figure 8), where the highest DOTA-Gd concentrations are found near the injection site (the cisterna magna), in line with previous reports (Iliff et al., 2012). In the *Mlc1* KO model, dispersion of DOTA-Gd is presumably affected in the SAS and the parenchyma.

With regard to tracer dispersion in the SAS and the cisternae, our anatomical MRI showed that the brain volume is greater in the *Mlc1* KO mouse than in the WT (see Figure 1). These variations in the geometry of the SAS may account for much of the difference between the *Mlc1* KO mice and WT mice. Although tracer concentrations appear to be similar in the cerebellum (close to the injection site), they are much lower in the more distant septal area of *Mlc1* KO mice – suggesting that tracer transport within the SAS is restricted.

With regard to parenchymal dispersion, we showed that MLC1 is essential for the position of the astrocytes’ perivascular endfeet. Thus, in *Mlc1* KO mice, the formation of the perivascular space (as a conduit for solute distribution) is likely to be deficient. This aspect is revealed by the slope of the tracer’s concentration-time curve, which indicate slower kinetics in *Mlc1* KO mice; this might be due to poor integrity of the perivascular space. The higher volume of fluid in the *Mlc1* KO parenchyma (reflected by the increased apparent diffusion coefficient (ADC); Figure 1 and S1) might also be involved in this phenotype.

Would these defects be originated by the reduced perivascular astrocyte coverage or the reduced vasoconstriction itself? Addressing these questions will strengthen the potential implication of Mlc1 in the circulation and clearance of CSF.

The heart beat is the main driver of CSF circulation in the perivascular space (Iliff et al., 2013). The heart rate is very rapid and so the heart exerts a much greater driving force on the CSF than the vasodilation of the vessels induced by neuronal activity. Alterations in vascular contractility observed in *Mlc1* KO mice might be involved in the impaired CSF flux but this is unlikely.

All these points are now discussed in the revised version of the manuscript.

Iliff JJ, Lee H, Yu M, Feng T, Logan J, Nedergaard M, and Benveniste H. 2013. Brain-wide pathway for waste clearance captured by contrast-enhanced MRI. Journal of Clinical Investigation 123: 12991309.10.1172/jci67677.

Iliff JJ, Wang M, Liao Y, Plogg BA, Peng W, Gundersen GA, Benveniste H, Vates GE, Deane R, Goldman SA, et al., 2012. A paravascular pathway facilitates CSF flow through the brain parenchyma and the clearance of interstitial solutes, including amyloid β. Sci Transl Med 4: 147ra111.10.1126/scitranslmed.3003748.

3) The defective polarity of astrocytes should be better described by using other markers other than GFAP. The distribution of Aquaporin4, Cx43 or several glutamate transporters in the specific compartment of astrocytes can be examined.

GFAP is the marker typically used to analyze the astrocytes’ overall morphology and polarity. Nevertheless, we agree that it is of interest to study the molecular polarity of PvAPs. Indeed, morphological changes in the PvAPs and astrocytes and changes in polarity in *Mlc1* KO might all influence the localization of molecules in PvAPs. To address this question, we performed a quantitative stimulated emission depletion (STED) analysis of protein localization in PvAPs. Our results indicate that the perivascular localization of aquaporin 4 was not affected. However, the density and size of Cx43 puncta were greater – indicating that the gap junctions in PvAPs are not organized in the same way in the *Mlc1* KO as in the WT. This observation is consistent with our electron microscopy observations of perivascular astrocytic processes stacked on the top of each other and linked by extended gap junctions.

We have also added results for Kir4.1, a potassium channel that is expressed preferentially in PvAPs. The Kir4.1 expression level in *Mlc1* KO was lower at all stages of development, indicating that perivascular potassium homeostasis was probably perturbed. These results are interesting because (i) epilepsy is a significant component of megalencephalic leukoencephalopathy (Dubey et al., 2018; Yalcinkaya et al., 2003), and (ii) Kir4.1 deletion or downregulation is associated with greater susceptibility to epilepsy (Sibille et al., 2014). These points are now discussed.

Dubey M, Brouwers E, Hamilton EMC, Stiedl O, Bugiani M, Koch H, Kole MHP, Boschert U, Wykes RC, Mansvelder HD, et al., 2018. Seizures and disturbed brain potassium dynamics in the leukodystrophy megalencephalic leukoencephalopathy with subcortical cysts. Ann Neurol 83: 636649.10.1002/ana.25190.

Sibille J, Pannasch U, and Rouach N. 2014. Astroglial potassium clearance contributes to short-term plasticity of synaptically evoked currents at the tripartite synapse. J Physiol 592: 87-102.jphysiol.2013.261735 [pii] 10.1113/jphysiol.2013.261735.

Yalcinkaya C, Yuksel A, Comu S, Kilic G, Cokar O, and Dervent A. 2003. Epilepsy in vacuolating megalencephalic leukoencephalopathy with subcortical cysts. Seizure 12: 388-396.10.1016/s10591311(02)00350-3.

Reviewer #1 (Recommendations for the authors):1. In general, Western blot should be presented with control loading.

We used Histone 3 or BioRad’s Stainfree acrylamide gels to normalize our Western blot results. We now provide all the Western blots and the corresponding Histone 3 and Stainfree images as a supplementary document. However, we do not feel that it is necessary to show all this information in the figures.

2. Figure 5 and 8, as well as Figure 6 and 9 are almost identical. I suggest either combining these figures into two main figures or send the mirror figures to supplementary figures.

We have modified the figures accordingly.

3. In Figure 4B, why Glialcam western showed non-significant changes at P15 and P60 although IHC data showed Glialcam is not expressed at all in MLlc1 KO astrocytes?

MLC1 is required for the targeting of GlialCAM to the astrocytic membrane. Diffuse GlialCAM in absence of MLC1 is hardly detectable in situ, while the protein is still detected in a Western blot. This has been demonstrated previously (Hoegg-Beiler et al., 2014).

Hoegg-Beiler MB, Sirisi S, Orozco IJ, Ferrer I, Hohensee S, Auberson M, Godde K, Vilches C, de Heredia ML, Nunes V*, et al.,* 2014. Disrupting MLC1 and GlialCAM and ClC-2 interactions in leukodystrophy entails glial chloride channel dysfunction. *Nat Commun **5***: 3475.ncomms4475 [pii] 10.1038/ncomms4475.

4. In Figure 4C, Aquaporin4 western blot seems to be just copied from the previous western blot data from Figure 4B.

Yes, they are the same Western blots; we had duplicated them to make the figure easier to understanding. We have now removed this duplication.

Reviewer #2 (Recommendations for the authors):This very interesting manuscript by Gilbert and colleagues uncovers that the astrocyte specific membrane protein MLC1, the mutation of which causes a rare disease called megalencephalic leukoencephalopathy with subcortical Cysts (MLC), plays a fundamental role in the postnatal development of the gliovascular unit and the organization of the perivascular astrocyte processes, in particular. To reach this conclusion, the authors used an elegant multiscale approach including in vivo MRI, in vivo functional ultrasound, ex vivo analysis of vascular constriction, anatomical approaches at the light and electron microscopic level, and molecular characterization of the gliovascular unit from isolated microvessels. The manuscript is very well-written although it uses too many (unnecessary) abbreviations, which prevents a fluid reading of the manuscript, results are well illustrated and convincing and the discussion is reasonable. The paper certainly is an important piece of work, which would feature well in eLife.I have a major concern regarding the results reported in Figure 4D, which seem somewhat contradictory to those shown in Figure 6A-F. Indeed, the authors report in Figure 4D that there is less Neurofilament-M protein around isolated microvessels in MLC1 KO mice, whereas Figure 6A-F shows that these animals have more neuronal processes in contact with the vessels than in wiltypes. How can the authors explain this?

The two situations are not comparable. On one hand, we observed the structure of the gliovascular unit in situ in fixed tissues. On the other, we mechanically purified microvessels. The detachment of astrocytic processes and associated neuronal fibers (linked to the mechanical dissociation of microvessels in the *Mlc1* KO mouse) was clearly not counterbalanced by the presence of neuronal fibers contacting the vessels.

Reviewer #3 (Recommendations for the authors):Astrocytes unique to the central nervous system such as the brain is known to play a role in maintaining BBB integrity by surrounding brain vessels tightly vis their end-feet structure. However, how astrocytes form end-feet along brain vessels is largely unknown. The present manuscript by Gilbert et al., describes how perivascular astrocytic processes are established during postnatal development and pinpoint a player involved. They find that MLC1 deletion resulted in disorganized perivascular astrocytic processes and defective contractility of vascular smooth muscle cells, thereby leading to impaired neurovascular coupling and intraparenchymal fluid clearance. To assess the functions on top of anatomy, they monitored blood perfusion and fluid transport of CSF in vivo using ultrasound imaging and MRI. I think that only a few issues remain to be clarified publication of this manuscript.1) In Figure 7D and 7G, but not in 7E and 7F, the basal values are significantly high in Mlc1 KO mice at 10 min after Gd infusion. How is the pattern of basal Gd concentration different depending on the brain area?

Paravascular transport (as revealed by the injection of a tracer into the CSF) depends mainly on dispersion of the tracer in the subarachnoid space (SAS), the cisternae, and the parenchyma (including the interstitial and perivascular spaces). The uneven, slow dispersion of the tracer within the SAS (compared with dispersion in the blood) means that the tracer’s kinetics in the parenchyma are regiondependent. These differences can be accentuated by regional differences in the anatomy of the brain’s vasculature, i.e. the presence or absence of a perivascular space and the vessel’s topology. Lastly, the amount of DOTA-Gd available for diffusion within the parenchyma depends directly on its local concentration in the SAS. This can be seen on our contrast concentration maps (see Figure 8), where the highest DOTA-Gd concentrations are found near the injection site (the cisterna magna), in line with previous reports (Iliff et al., 2012).

Iliff JJ, Wang M, Liao Y, Plogg BA, Peng W, Gundersen GA, Benveniste H, Vates GE, Deane R, Goldman SA*, et al.,* 2012. A paravascular pathway facilitates CSF flow through the brain parenchyma and the clearance of interstitial solutes, including amyloid β. *Sci Transl Med* 4: 147ra111.10.1126/scitranslmed.3003748.

2) Although endothelial cells look normal in immunofluorescence images and TEM images, it needs to be clarified whether the function of BBB is affected by assessing leakage of tracers infused into blood.

We provide (in Figure 1D) an extensive analysis of the BBB’s integrity vs. the in situ perfusion of radiolabeled sucrose. This 30-year-old in vivo technique is extensively used in the BBB field to measure the qualitative and quantitative solute kinetic properties of the BBB and to assess the BBB’s physical integrity with compounds that normally fail to cross the BBB (Takasato et al., 1984). Here, we used sucrose – one of the best low-molecular-weight markers of vascular integrity. This hydrophilic compound does not bind to plasma proteins. Moreover, sucrose has no dedicated transporters in mammals and thus barely cross the BBB/cells (Takasato et al., 1984). Our data demonstrate that the BBB is not leaky in *Mlc1* KO – even under vascular shear stress conditions. Since astrocytes are critical for the maintenance of BBB integrity, our results indicate that the morphological change in the GVU induced by the absence of MLC1 is not enough to perturb BBB integrity. This point is now discussed in the revised version of the manuscript.

Takasato Y, Rapoport SI, and Smith QR. 1984. An in situ brain perfusion technique to study cerebrovascular transport in the rat. Am J Physiol 247: H484-493